# Inductive Logical Query Answering in Knowledge Graphs

**Mikhail Galkin**
Mila, McGill University
mikhail.galkin@mila.quebec

**Zhaocheng Zhu**
Mila, Université de Montréal
zhuzhaoc@mila.quebec

**Hongyu Ren**
Stanford University
hyren@stanford.edu

**Jian Tang**
Mila, HEC Montréal, CIFAR AI Chair
jian.tang@hec.ca

## Abstract

Formulating and answering logical queries is a standard communication interface for knowledge graphs (KGs). Alleviating the notorious incompleteness of real-world KGs, neural methods achieved impressive results in link prediction and complex query answering tasks by learning representations of entities, relations, and queries. Still, most existing query answering methods rely on transductive entity embeddings and cannot generalize to KGs containing new entities without retraining the entity embeddings. In this work, we study the inductive query answering task where inference is performed on a graph containing new entities with queries over both seen and unseen entities. To this end, we devise two mechanisms leveraging inductive *node* and *relational structure* representations powered by graph neural networks (GNNs). Experimentally, we show that inductive models are able to perform logical reasoning at inference time over unseen nodes generalizing to graphs up to 500% larger than training ones. Exploring the efficiency–effectiveness trade-off, we find the inductive *relational structure* representation method generally achieves higher performance, while the inductive *node representation* method is able to answer complex queries in the *inference-only* regime without any training on queries and scales to graphs of millions of nodes. Code is available at https://github.com/DeepGraphLearning/InductiveQE.

## 1 Introduction

Traditionally, querying knowledge graphs (KGs) is performed via databases using structured query languages like SPARQL. Databases can answer complex queries relatively fast under the assumption of *completeness*, i.e., there is no missing information in the graph. In practice, however, KGs are notoriously incomplete [32]. Embedding-based methods that learn vector representations of entities and relations are known to be effective in *simple link prediction* predicting heads or tails of query patterns *(head, relation, ?)*, e.g., *(Einstein, graduate, ?)*, as common in *KG completion* [1, 17].

Complex queries are graph patterns expressed in a subset of first-order logic (FOL) with operators such as intersection ($\wedge$), union ($\vee$), negation ($\neg$) and existentially quantified ($\exists$) variables[1], e.g., $?U.\exists V :$ Win(NobelPrize, $V$) $\wedge$ Citizen(USA, $V$) $\wedge$ Graduate($V, U$) (Fig. 1). Complex queries define a superset of KG completion. The conventional KG completion (link prediction) task can be viewed as a complex query with a single triplet pattern without logical operators, e.g., Citizen(USA, $V$), which we also denote as a *projection* query.

---

[1]The universal quantifier ($\forall$) is often discarded as in real-world KGs there is no node connected to all others.

36th Conference on Neural Information Processing Systems (NeurIPS 2022).

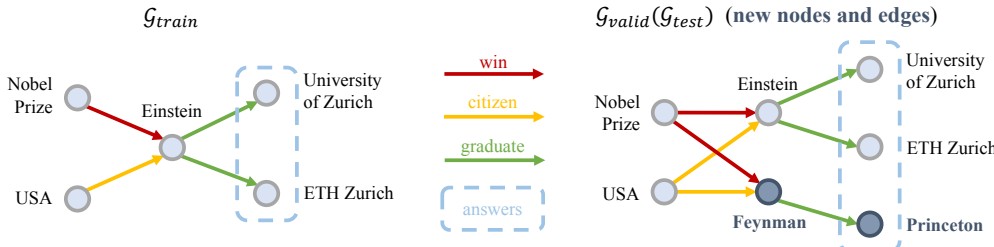

**Where did US citizens with Nobel Prize graduate?** $q = v. \exists u: Win(Nobel\ Prize, u) \land Citizen(USA, u) \land Graduate(u, v)$

Figure 1: Inductive query answering problem: at inference time, the graph is updated with new nodes `Feynman` and `Princeton` and edges such that the same query now has more answers.

To tackle complex queries on incomplete knowledge graphs, *query embedding* methods are proposed to execute logic operations in the latent space, including variants that employ geometric [15, 23, 38], probabilistic [24, 9], neural-symbolic [26, 8, 5], neural [21, 4], and GNN [11, 3] approaches for learning entity, relation, and query representations.

However, this very fact of learning a separate embedding for each entity makes those methods inherently *transductive* i.e., they are bound to the space of learned entities and cannot generalize to unseen entities without retraining the whole embedding matrix which can be prohibitively expensive in large graphs. The problem is illustrated in Fig. 1: given a graph about `Einstein` and a logical query *Where did US citizens with Nobel Prize graduate?*, transductive QE methods learn to execute logical operators and return the answer set {`University of Zurich, ETH Zurich`}. Then, the graph is updated with new nodes and edges about `Feynman` and `Princeton`, and the same query now has more correct answers {`University of Zurich, ETH Zurich, Princeton`} as new unseen entities satisfy the query as well.

Such *inductive inference* is not possible for transductive models as they do not have representations for new `Feynman` and `Princeton` nodes. In the extreme case, inference graphs might be disconnected from the training one and only share the set of relations. Therefore, inductive capabilities are a key factor to transferring trained query answering models onto updated or entirely new KGs.

In this work, we study answering complex queries in the inductive setting, where the model has to deal with unseen entities at inference time. Inspired by recent advancement in inductive Kg completion [42, 13], we devise two solutions for learning inductive representations for complex query: 1) The first solution, NodePiece-QE, extends the inductive node representation model NodePiece [13] to complex query answering. NodePiece-QE learns **inductive representations of each entity** as a function of tokens from a fixed-size vocabulary, and answers complex query with a non-parametric logical query executor [5]. The advantages of NodePiece-QE are that it only needs to be trained on simple link prediction data, answers complex queries in the *inference-only* mode, and can scale to large KGs. 2) The second solution, GNN-QE [40], extends the inductive KG completion model NBFNet [42] for complex query answering. Originally, GNN-QE was studied only in the transductive setting. Here, we analyze its inductive capabilities. GNN-QE learns **inductive representations of the relational structure** without entity embeddings, and uses the relational structure between the query constants and the answers to make the prediction. GNN-QE can be trained end-to-end on complex queries, achieves much better performance than NodePiece-QE, but struggles to scale to large KGs.

To the best of our knowledge, this is the first work to study complex logical query answering in the inductive setting *without any additional features like entity types or textual descriptions*. Conducting experiments on a novel benchmarking suite of 10 datasets, we find that 1) both inductive solutions exhibit non-trivial performance answering logical queries over unseen entities and query patterns; 2) inductive models demonstrate out-of-distribution generalization capabilities to graphs up to 500% larger than training ones; 3) akin to updatable databases, inductive methods can successfully find new correct answers to known training queries after adding new nodes and edges; 4) the inductive *node representation* method scales to answering logical queries over a graph of 2M nodes with 500k new unseen nodes; 5) GNN-based models still exhibit some difficulties [20, 35] generalizing to graphs larger than those they were originally trained on.

## 2 Related Work

**Knowledge Graph Completion.** Knowledge graph completion, a.k.a. simple link prediction, has been widely studied in the *transductive* paradigm [6, 33, 27, 37], i.e., when training and inference are performed on the same graph with a fixed set of entities. Generally, these methods learn a shallow embedding vector for each entity. We refer the audience to respective surveys [1, 17] covering dozens of transductive embedding methods. The emergence of message passing [14] and graph neural networks (GNNs) has led to more advanced, *inductive* representation learning approaches that model entity or triplet representations as a function of the graph structure in its neighborhood. GraIL [28] learns triplet representations based on the subgraph structure surrounding the two entities. NeuralLP [34], DRUM [25] and NBFNet [42] learn the pairwise entity representations based on the set of relation paths between two entities. NodePiece [13] learns entity representations from a fixed-size vocabulary of tokens that can be anchor nodes in a graph or relation types.

**Complex Query Answering.** In the complex (multi-hop) query answering setup with logical operators, existing models employ different approaches, e.g., geometric [15, 23, 38], probabilistic [24, 9], neural-symbolic [26, 8, 5], neural [21, 4], and GNN [11, 3]. Still, all the approaches are created and evaluated exclusively in the transductive mode where the set of entities does not change at inference time. To the best of our knowledge, there is no related work in inductive logical query answering when the inference graph contains new entities. With our work, we aim to bridge this gap and extend inductive representation learning algorithms to logical query answering. In particular, we focus on the inductive setup where an inference graph is a superset of a training graph[2] such that 1) inference queries require reasoning over both seen and new entities; 2) original training queries might have more correct answers at inference time with the addition of new entities.

## 3 Preliminaries and Problem Definition

**Knowledge Graph and Inductive Setup.** Given a finite set of entities $\mathcal{E}$, a finite set of relations $\mathcal{R}$, and a set of triples (edges) $\mathcal{T} = (\mathcal{E} \times \mathcal{R} \times \mathcal{E})$, a knowledge graph $\mathcal{G}$ is defined as $\mathcal{G} = (\mathcal{E}, \mathcal{R}, \mathcal{T})$. Accounting for the inductive setup, we define a *training* graph $\mathcal{G}_{train} = (\mathcal{E}_{train}, \mathcal{R}, \mathcal{T}_{train})$ and an *inference* graph $\mathcal{G}_{inf} = (\mathcal{E}_{inf}, \mathcal{R}, \mathcal{T}_{inf})$ such that $\mathcal{E}_{train} \subset \mathcal{E}_{inf}$ and $\mathcal{T}_{train} \subset \mathcal{T}_{inf}$. That is, the *inference* graph extends the training graph with new entities and edges[3]. The inference graph $\mathcal{G}_{inf}$ is an incomplete part of the not observable complete graph $\hat{\mathcal{G}}_{inf} = (\mathcal{E}_{inf}, \mathcal{R}, \hat{\mathcal{T}}_{inf})$ with $\hat{\mathcal{T}}_{inf} = \mathcal{T}_{inf} \cup \mathcal{T}_{pred}$ whose missing triples $\mathcal{T}_{pred}$ have to be predicted at inference time.

**First-Order Logic Queries.** Applied to KGs, a first-order logic (FOL) query $\mathcal{Q}$ is a formula that consists of constants $\mathcal{C}$ ($\mathcal{C} \subseteq \mathcal{E}$), variables $\mathcal{V}$ ($\mathcal{V} \subseteq \mathcal{E}$, existentially quantified), relation *projections* $R(a, b)$ denoting a binary function over constants or variables, and logic symbols ($\exists, \wedge, \vee, \neg$). The answers $A_{\mathcal{G}}(\mathcal{Q})$ to the query $\mathcal{Q}$ are assignments of variables in a formula such that the instantiated query formula is a subgraph of the complete graph $\hat{\mathcal{G}}$.

Fig. 1 illustrates the logical form of a query *Where did US citizens with Nobel Prize graduate?* as $?U.\exists V : \mathtt{Win}(\mathtt{NobelPrize}, V) \wedge \mathtt{Citizen}(\mathtt{USA}, V) \wedge \mathtt{Graduate}(V, U)$ where $\mathtt{NobelPrize}$ and $\mathtt{USA}$ are *constants*; $\mathtt{Win}, \mathtt{Citizen}, \mathtt{Graduate}$ are *relation projections* (labeled edges); $V, U$ - *variables* such that $V$ is an existentially quantified free variable and $U$ is the projected bound *target* variable of the query. Common for the literature, we aim at predicting assignments of the query *target* whereas assignments of intermediate variables might not always be explicitly interpreted depending on the model architecture. In the example, the answer set $A_{\mathcal{G}}(\mathcal{Q})$ is a binding of a target variable $U$ to constants $\mathtt{University\ of\ Zurich}$ and $\mathtt{ETH\ Zurich}$.

**Inductive FOL Queries.** In the standard transductive query answering setup, query constants and variables at both training and inference time belong to the same set of entities, i.e., $\mathcal{C}_{train} = \mathcal{C}_{inf} \subseteq \mathcal{E}, \mathcal{V}_{train} = \mathcal{V}_{inf} \subseteq \mathcal{E}$. In the inductive setup covered in this work, query constants and variables at inference time belong to a different and larger set of entities $\mathcal{E}_{inf}$ from the inference graph $\mathcal{G}_{inf}$, i.e., $\mathcal{C}_{train} \subseteq \mathcal{E}_{train}, \mathcal{V}_{train} \subseteq \mathcal{E}_{train}$ but $\mathcal{C}_{inf} \subseteq \mathcal{E}_{inf}, \mathcal{V}_{inf} \subseteq \mathcal{E}_{inf}$. This also leads to the fact that training queries executed over the inference graph might have more correct answers, i.e., $A_{\mathcal{G}_{train}}(\mathcal{Q}) \subseteq A_{\mathcal{G}_{inf}}(\mathcal{Q})$. For example (cf. Fig. 1), the inference graph is updated with new nodes $\mathtt{Feynman}$, $\mathtt{Princeton}$ and their

---

[2]The set of relation types is fixed.

[3]Note that the set of relation types $\mathcal{R}$ remains the same.

new respective edges. The same query now has a larger set of intermediate variables satisfying the formula (`Feynman`) and an additional correct answer `Princeton`. Therefore, inductive generalization is essential for obtaining representations of such new nodes and enabling logical reasoning over both seen and new nodes, i.e., finding more answers to known queries in larger graphs or answering new queries with new constants. In the following section, we describe two approaches for achieving inductive generalization with different parameterization strategies.

## 4 Method

**Inductive Representations of Complex Queries.** Given a complex query $\mathcal{Q} = (\mathcal{C}, \mathcal{R}_{\mathcal{Q}}, \mathcal{G})$, the goal is to rank all possible entities according to the query. From a representation learning perspective, this requires us to learn a conditional representation function $f(e|\mathcal{C}, \mathcal{R}_{\mathcal{Q}}, \mathcal{G})$ for each entity $e \in \mathcal{E}$. Transductive methods learn a shallow embedding for each answer entity $e \in \mathcal{E}$, and, therefore, cannot generalize to unseen entities. For inductive methods, the function $f(e|\mathcal{C}, \mathcal{R}_{\mathcal{Q}}, \mathcal{G})$ should generalize to some unseen answer entity $e'$ (or unseen constant entity $c' \in \mathcal{C}'$) at inference time. Here, we discuss two solutions for devising such an inductive function.

The first solution is to **parameterize the representation of each entity** $e$ **as a function of an invariant vocabulary** of *tokens* that does not change at training and inference. Particularly, the vocabulary might consist of unique relation types $\mathcal{R}$ that are always the same for $\mathcal{G}_{train}$ and $\mathcal{G}_{inf}$, and we are able to infer the representation of an unseen answer entity (or an unseen constant entity) as a function of its incident relations (cf. Fig. 2 left). The idea has been studied in NodePiece [13] for simple link prediction. Here, we adopt a similar idea to learn inductive entity representations for complex query answering. Once we obtain the representations for unseen entities, we can use any off-the-shelf decoding method (e.g., CQD-Beam [5]) for predicting the answer to the complex query. We denote this strategy as NodePiece-QE.

The second solution is to **parameterize** $f(e|\mathcal{C}, \mathcal{R}_{\mathcal{Q}}, \mathcal{G})$ **as a function of the relational structure**. Intuitively, an answer of a complex query can be decided solely based on the relational structure between the query constants and the answer (Fig. 1). Even after anonymizing entity names (and, hence, not learning any explicit entity embedding), we can still infer `Princeton` as an answer since it forms a distinctive relational structure ⤳ with the query constants and conforms to the query structure. Similarly, intermediate nodes will be deemed correct if they follow a relational structure ⟩. In other words, we do not need to know the answer node is `Princeton`, but only need to know the relative position of `Princeton` w.r.t. the constants like `Nobel Prize` and `USA`. Based on this idea, we design $f(e|\mathcal{C}, \mathcal{R}_{\mathcal{Q}}, \mathcal{G})$ to be a relational structure search function. Such an idea has been studied in Neural Bellman-Ford Networks (NBFNet) [42] to search for a single relation in simple link prediction. Applied to complex queries, GNN-QE [40] chains several NBFNet instances with differentiable logic operations to learn inductive complex query in an end-to-end fashion. So far, GNN-QE was evaluated solely on transductive tasks. Here we extend it to the inductive setup.

### 4.1 NodePiece-QE: Inductive Node Representation

Here we aim at reconstructing node representations for seen and unseen entities without learning shallow node embedding vectors. To this end, we employ NodePiece [13], a compositional tokenization approach that learns an invariant vocabulary of *tokens* shared between training and inference graphs. Formally, given a vocabulary of tokens $t_i \in T$, each entity $e_i$ is deterministically hashed into a set of representative tokens $e_i = [t_1, \ldots, t_k]$. An entity vector $\boldsymbol{e}_i$ is then obtained as a function of token embeddings $\boldsymbol{e}_i = f_\theta([\boldsymbol{t}_i, \ldots, \boldsymbol{t}_k]), \boldsymbol{t}_i \in \mathbb{R}^d$ where the encoder function $f_\theta : \mathbb{R}^{k \times d} \to \mathbb{R}^d$ is parameterized with a neural network $\theta$.

Since the set of relation types $\mathcal{R}$ is invariant for training and inference graphs, we can learn relation embeddings $\boldsymbol{R} \in \mathbb{R}^{|\mathcal{R}| \times d}$ and our vocabulary of learnable tokens $T$ is comprised of distinct relation types such that entities are hashed into a set of unique incident relation types. For example (cf. Fig. 2 left), a middle node from a training graph $\mathcal{G}_{train}$ is hashed with a set of relations $e_i = [⇊↑]$ that stands for two unique incoming relations ⇊ and one unique outgoing relation ↑. Passing the hashes through $f_\theta$, we can reconstruct the whole entity embedding matrix $\boldsymbol{E} \in \mathbb{R}^{|\mathcal{E}_{train}| \times d}$. Additionally, it is possible to enrich entity and relation embeddings by passing them through a relational GNN encoder [31] over a target graph $\mathcal{G}$: $\boldsymbol{E}', \boldsymbol{R}' = \text{GNN}(\boldsymbol{E}, \boldsymbol{R}, \mathcal{G})$. In both ways, the entity embedding matrix $\boldsymbol{E}$ encodes a *joint* probability distribution $p(h, r, t)$ for all triples in a graph.

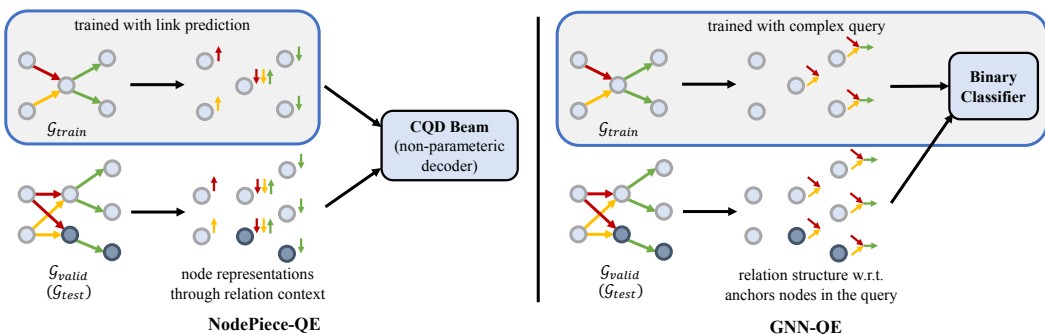

Figure 2: Inductive node representation (NodePiece-QE, left) and relational structure (GNN-QE, right) strategies for complex logical query answering. In NodePiece-QE, we obtain inductive node representations through the invariant set of tokens (here, through incident relation types). NodePiece-QE is an inference-only approach, pre-trained with simple *1p* link prediction and can be directly applied to inductive complex queries with a non-parametric decoder (e.g., CQD Beam). In GNN-QE, we learn the the relative structure of each node w.r.t. the anchor nodes in the query. GNN-QE is trainable end-to-end on *complex queries*.

Having a uniform featurization mechanism for both seen and unseen entities, it is now possible to apply any previously-transductive complex query answering model with learnable entity embeddings and logical operators [23, 11, 24, 8]. Moreover, it was recently shown [5] that a combination of simple link prediction pre-training and a non-parametric logical executor allows to effectively answer complex FOL queries in the *inference-only* regime without training on any complex query sample. We adopt this Continuous Query Decomposition algorithm with beam search (CQD-Beam) as the main query answering decoder. CQD-Beam relies only on entity and relation embeddings $\boldsymbol{E}, \boldsymbol{R}$ pre-trained on a simple *1p* link prediction task. Then, given a complex query, CQD-Beam applies *t-norms* and *t-conorms* [19] that execute conjunctions ($\wedge$) and disjunctions ($\vee$) as non-parametric algebraic operations in the embedding space, respectively.

In our inductive setup (Fig. 2), we train a NodePiece encoder $f_\theta$ and relation embeddings $\boldsymbol{R}$ (and optionally a GNN) on the *1p* link prediction task over the training graph $\mathcal{G}_{train}$. We then apply the learned encoder to materialize entity representations of the inference graph $\boldsymbol{E} \in \mathbb{R}^{|\mathcal{E}_{inf}| \times d}$ and send them to CQD-Beam that performs a non-parametric decoding of complex FOL queries over new unseen entities. The inference-only nature of NodePiece-QE is designed to probe the abilities for zero-shot generalization in performing complex logical reasoning over larger graphs.

### 4.2 GNN-QE: Inductive Relational Structure Representation

The second strategy relies on learning inductive relational structure representations instead of explicit node representations. Having the same set of relation types $\mathcal{R}$ at training and inference time, we can parameterize each entity based on the relative relational structure between it and the anchor nodes in a given query. For instance (Fig. 2 right), given a query with a particular relational structure ⤳ and a set of anchor nodes, the representation of each node captures its relational structure relative to the anchor nodes. Each neighborhood expansion step is equivalent to a *projection* step in a complex query. In our example, immediate neighboring nodes will capture the intersection pattern ⌄ , and further nodes, in turn, capture the extended *intersection-projection* structure ⤳ .

Therefore, a node is likely to be an answer if its captured (or predicted) relational structure conforms with the query relational structure. As long as the set of relations is fixed, relation *projection* is performed in the same way for training or new unseen nodes. The idea of a one-hop (*1p*) projection for simple link prediction has been proposed by Neural Bellman-Ford Networks (NBFNet) [42].

In particular, given a relation *projection* query $(h, r, ?)$, NBFNet assigns unique initial states $\boldsymbol{h}^{(0)}$ to all nodes in a graph by applying an indicator function $\boldsymbol{h}^{(0)} = \text{INDICATOR}(h, v, r)$, i.e., a head node $h$ is initialized with a learnable relation embedding $\boldsymbol{r}$ and all other nodes are initialized with zeros. Then, NBFNet applies $L$ relational message passing GNN layers where each layer $l$ has its own learnable relation embedding matrix $\boldsymbol{R}^{(l)}$ obtained as a projection (and reshaping) of thein initial relation

$\boldsymbol{R}^{(l)} = \boldsymbol{W}^{(l)}\boldsymbol{r} + \boldsymbol{b}^{(l)}$. Final layer representations $\boldsymbol{h}^{(L)}$ are passed through an MLP and the sigmoid function $\sigma$ to get a probability distribution over all nodes in a graph $p(t|h, r) = \sigma(\text{MLP}(\boldsymbol{h}^{(L)}))$. As each projection query spawns a uniquely initialized graph and message passing procedure, NBFNet is seen to be applying a *labeling trick* [36] to model a conditional probability distribution $p(t|h, r)$ that is provably more expressive than a joint distribution $p(h, r, t)$ produced by standard graph encoders.

Applied to complex queries, chaining $k$ NBFNet instances allows us to answer $k$-hop projection queries, e.g., two instances for *2p* queries. GNN-QE employs NBFNet as a *trainable* projection operator and endows it with differentiable, non-parametric *product* logic for modeling conjunction ($\wedge$), disjunction ($\vee$), and negation ($\neg$) over the *fuzzy sets* of all entities $\boldsymbol{x} \in [0, 1]^{\mathcal{E}}$, i.e., after applying a logical operator (discussed in Appendix A), each entity's degree of truth is associated with a scalar in range $[0, 1]$. For the $i$-th hop projection, the indicator function initializes a node state $\boldsymbol{h}_e^{(0)}$ with a relation vector $\boldsymbol{r}_i$ weighted by a scalar probability predicted in the previous hop $x_e$: $\boldsymbol{h}_e^{(0)} = x_e \boldsymbol{r}_i$. Differentiable logical operators allow training GNN-QE end-to-end on complex queries.

## 5 Experiments

We designed the experimental agenda to demonstrate that inductive representation strategies are able to: 1) answer complex logical queries over new, unseen entities at inference time, i.e., when query anchors are new nodes (Section 5.2); 2) predict new correct answers for known *training* queries when executed over larger inference graphs, i.e., when query anchors come from the training graph but variables and answers belong to the larger inference graph (Section 5.3); 3) generalize to inference graphs of up to 500% larger than training graphs; 4) scale to inductive query answering over graphs of millions of nodes when updated with 500k new nodes and 5M new edges (Section 5.5).

### 5.1 Setup & Dataset

**Dataset.** Due to the absence of inductive logical query benchmarks, we create a novel suite of datasets based on FB15k-237 [29] (open license) and following the query generation process of BetaE [24]. Given a source graph with $\mathcal{E}$ entities, we sample $|\mathcal{E}_{train}| = r \cdot |\mathcal{E}|, r \in [0.1, 0.9]$ nodes to induce a training graph $\mathcal{G}_{train}$. For validation and test graphs, we split the remaining set of entities into two non-overlapping sets each with $\frac{1-r}{2}|\mathcal{E}|$ nodes. We then merge training and unseen nodes into the inference set of nodes $\mathcal{E}_{inf}$ and induce inference graphs for validation and test from those sets, respectively, i.e., $\mathcal{E}_{inf}^{val} = \mathcal{E}_{train} \cup \mathcal{E}_{val}$ and $\mathcal{E}_{inf}^{test} = \mathcal{E}_{train} \cup \mathcal{E}_{test}$. That is, validation and test inference graphs both extend the training graph but their sets of new entities are disjoint. Finally, we sample and remove 15% of edges $\mathcal{T}_{pred}$ in the inference graphs as missing edges for sampling queries with those missing edges. Overall, we sample 9 such datasets based on different choices of $r$, which result in the ratios of inference graph size to the training graph $\mathcal{E}_{inf}/\mathcal{E}_{train}$ from 106% to 550%.

For each dataset, we employ the query sampler from BetaE [24] to extract 14 typical query types *1p/2p/3p/2i/3i/ip/pi/2u/up/2in/3in/inp/pin/pni*. Training queries are sampled from the training graph $\mathcal{G}_{train}$, validation and test queries are sampled from their respective inference graphs $\mathcal{G}_{inf}$ where at least one edge belongs to $\mathcal{T}_{pred}$ and has to be predicted at inference time.

As inference graphs extend training graphs, training queries are very likely to have new answers when executed over $\mathcal{G}_{inf}$ with simple graph traversal and without any link prediction. We create an additional set of true answers for all training queries executed over the test inference graph $\mathcal{G}_{inf}^{test}$ to measure the entailment capabilities of query answering models. This is designed to be an inference task and extends the *faithfullness* evaluation of [26]. Dataset statistics can be found in Appendix B.

**Evaluation Protocol.** Following the literature [24], query answers are separated into two sets: *easy answers* that only require graph traversal over existing edges, and *hard answers* that require inferring missing links to achieve the answer node. For the main experiment, evaluation involves ranking of *hard* answers against all entities having easy ones filtered out. For evaluating training queries on inference graphs, we only have *easy* answers and rank them against all entities. We report Hits@10 as the main performance metric on different query types.

**Implementation Details.** All NodePiece-based models [13] were pre-trained until convergence on a simple *1p* link prediction task with the relations-only vocabulary and entity tokenization, MLP encoder, and ComplEx [30] scoring function. We used a 2-layer CompGCN [31] as an

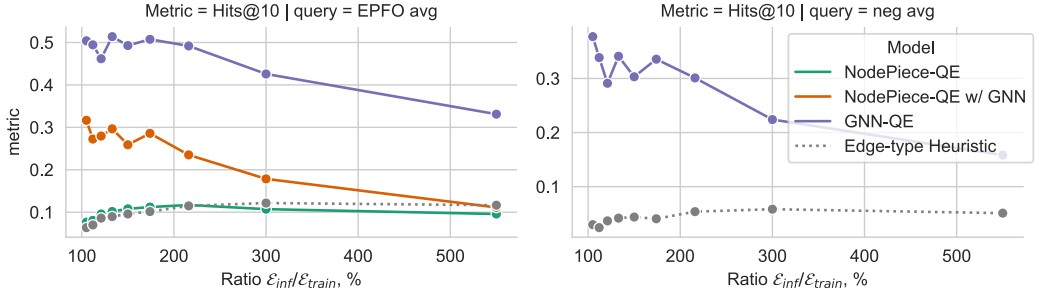

Figure 3: Aggregated Hits@10 performance of **test queries** (involving unseen entities) executed on inference graphs of different ratios compared to training graphs. NodePiece-based models are *inference-only* and support EPFO queries, GNN-QE is trainable and supports negation queries.

Table 1: Test Hits@10 results (%) on answering inductive FOL queries when $\mathcal{E}_{inf}/\mathcal{E}_{train} = 175\%$. $avg_p$ is the average on EPFO queries ($\wedge$, $\vee$). $avg_n$ is the average on queries with negation.

| Model | $avg_p$ | $avg_n$ | 1p | 2p | 3p | 2i | 3i | pi | ip | 2u | up | 2in | 3in | inp | pin | pni |
|---|---|---|---|---|---|---|---|---|---|---|---|---|---|---|---|---|
| | | | | | | *Transductive* | | | | | | | | | | |
| BetaE | 1.3 | 0.2 | 2.9 | 0.4 | 0.4 | 2.1 | 3.3 | 1.5 | 0.7 | 0.2 | 0.2 | 0.1 | 0.2 | 0.2 | 0.1 | 0.1 |
| | | | | | | *Inductive Inference-only* | | | | | | | | | | |
| Edge-type Heuristic | 10.1 | 4.1 | 17.7 | 8.2 | 9.9 | 10.7 | 13.0 | 9.8 | 8.2 | 5.3 | 8.5 | 2.6 | 2.9 | 8.4 | 3.8 | 2.7 |
| NodePiece-QE | 11.2 | - | 25.5 | 8.2 | 8.4 | 12.4 | 13.9 | 9.9 | 8.7 | 7.0 | 6.8 | - | - | - | - | - |
| NodePiece-QE w/ GNN | 28.6 | - | 45.9 | 19.2 | 11.5 | 39.9 | 48.8 | 29.4 | 22.6 | 25.3 | 14.6 | - | - | - | - | - |
| | | | | | | *Inductive Trainable* | | | | | | | | | | |
| GNN-QE | 50.7 | 33.6 | 65.4 | 36.3 | 31.6 | 73.8 | 84.3 | 56.5 | 41.5 | 39.3 | 28.0 | 33.3 | 46.4 | 29.2 | 24.9 | 34.0 |

optional message passing encoder on top of NodePiece features. The non-parametric CQD-Beam [5] decoder for answering complex queries is tuned for each query type based on the validation set of queries, most of the setups employ a *product t-norm*, sigmoid entity score normalization, and beam size of 32. Following the literature, the GNN-QE models [40] were trained on 10 query patterns (*1p/2p/3p/2i/3i/2in/3in/inp/pin/pni*) where *ip/pi/2u/up* are only seen at inference time. Each model employs a 4-layer NBFNet [42] as a trainable projection operator with DistMult [33] composition function and PNA [10] aggregation. Other logical operators ($\wedge$, $\vee$, $\neg$) are executed with the non-parametric *product t-norm* and *t-conorm*. Both NodePiece-QE and GNN-QE are implemented[4] with PyTorch [22] and trained with the Adam [18] optimizer. NodePiece-QE models were pre-trained and evaluated on a single Tesla V100 32 GB GPU whereas GNN-QE models were trained and evaluated on 4 Tesla V100 16GB. All hyperparameters are listed in Appendix D. To show that the proposed models are non-trivial, we compare them with an *Edge-type Heuristic* baseline (Appendix E), which selects all entities that satisfy the relations in the last hop of the query in $\mathcal{G}_{inf}$.

### 5.2 Complex Query Answering over Unseen Entities on Differently Sized Inference Graphs

First, we probe *inference-only* NodePiece-based embedding models and *trainable* GNN-QE in the inductive setup, i.e., query answering over unseen nodes requiring link prediction over unseen nodes. As a sanity check, we compare them to the Edge-type Heuristic and a transductive BetaE model [24] trained with standard hyperparameters (Appendix D) on the reference dataset (with ratio $\mathcal{E}_{inf}/\mathcal{E}_{train}$ of 175%) with randomly initialized embeddings for unseen nodes at inference time. Table 1 summarizes the results on the reference dataset while Fig. 3 illustrates a bigger picture on all datasets (we provide a detailed breakdown by query type for all splits in Appendix C). The experiment on the tranductive BetaE confirms that pure transductive models can not generalize to graphs with unseen nodes.

With inductive models, however, we observe that even inference-only models pre-trained solely on simple *1p* link prediction exhibit non-trivial performance in answering queries with unseen entities. Particularly, the inference-only *NodePiece with GNN* baseline exhibits better performance over all query types and inference graphs up to 300% larger than training graphs.

---

[4]Code and data are available at https://github.com/DeepGraphLearning/InductiveQE

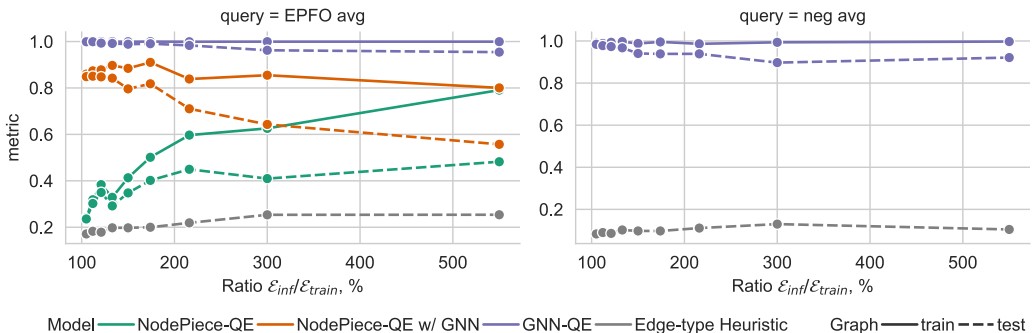

Figure 4: Aggregated Hits@10 performance of **training queries** on the original training and extended test inference graphs where queries have new correct answers. NodePiece-based models are *inference-only* and support EPFO queries, GNN-QE is trainable and supports negation queries.

The trainable GNN-QE models expectedly outperform non-trainable baselines and can tackle queries with negation ($\neg$). Here, we confirm that the *labeling trick* [36] and conditional $p(t|h, r)$ modeling better capture the relation projection problem than joint $p(h, r, t)$ encoding approaches.

Still, we notice that models with GNNs, i.e., inference-only NodePiece-QE with GNN and trainable GNN-QE, suffer from increasing the size of the inference graph and having more unseen entities. Reaching best results on $\mathcal{E}_{inf}/\mathcal{E}_{train}$ ratios around 130%, both approaches steadily deteriorate up until final 550% by 20 absolute Hits@10 points on EPFO queries and negation queries. We attribute this deterioration to the known generalization issues [20, 35] of message passing GNNs when performing inference over a larger graph than the network has seen during training. Recently, a few strategies have been proposed [7, 39] to alleviate this issue and we see it as a promising avenue for future work. On the other hand, a simple NodePiece-QE model without message passing retains similar performance independently of the inference graph size.

Lastly, we observe that lower performance of inference-only NodePiece models can be also attributed to underfitting (cf. train graph charts in Fig. 4). Although *1p* link predictors were trained until convergence (on the inductive validation set of missing triples), the performance of training queries on training graphs with *easy answers* that require only relation traversal without predicting missing edges is not yet saturated. This fact suggests that better fitting entity featurization (obtained by NodePiece or other strategies) could further improve the test performance in the inference-only regime. We leave the search of such strategies for future work.

## 5.3 Predicting New Answers for Training Queries on Larger Inference Graphs

Simulating the incremental addition of new edges in graph databases, we evaluate the performance of our inference-only and trainable QE models on *training* queries on the original training graph and extended inference graph (with added test edges). As databases are able to immediately retrieve new answers to known queries after updating the graph, we aim at exploring and quantifying this behaviour of neural reasoning models. In this experiment, we probe training queries and their *easy answers* that require performing only graph traversal without predicting missing links in the inference graph. While execution of training queries over the *training* graph indicates how well the model could fit training data, executing training queries over the bigger *inference* graph with new entities aims to capture basic reasoning capabilities of QE models in the inductive regime.

Particular challenges arising when executing training queries over a bigger graph are: (1) the same queries can have more correct answers as more new nodes and edges satisfying the query pattern might have been added (as in Fig. 1); (2) more new entities create a "distractor" setting with more false positives. Generally, evaluation of training queries on the inference graph can be considered as an extended version of the *faithfullness* [26] evaluation that captures how well a trained model can answer original training queries, i.e., memorization capacity. In all 9 datasets, most of training queries have at least one new correct answer in the inference graph (more details in Appendix B).

Fig. 4 illustrates the performance of the Edge-type Heuristic baseline, inference-only NodePiece-QE (without and with GNN) and trainable GNN-QE models. Generally, GNN-QE fits the training query data almost perfectly confirming the original finding [42] that NBFNet performs graph traversal akin to symbolic models. GNN-QE can also find new correct answers on graphs up to $300\%$ larger than training ones. Then, the performance deteriorates which we attribute to the *distractor* factor with more unseen entities and the mentioned generalization issue on larger inference graphs.

The inference-only NodePiece-QE models, as expected, do not fully fit the training data as they were never trained on complex queries. Still, the inference-only models exhibit non-trivial performance (compared to the Edge-type Heuristic) in finding more answers on graphs up to $200\%$ larger than training ones with relatively small performance margins compared to training queries. The most surprising observation is that GNN-free NodePiece-QE models improve the performance on both training and inference graphs as the graphs (and the $\mathcal{E}_{inf}/\mathcal{E}_{train}$ ratio) grow larger while GNN-enriched models steadily deteriorate. We attribute this growth to the relation-based NodePiece tokenization and its learned features that tend to be more discriminative in larger inference graphs where new nodes have smaller degree and thus can be better identified by their incident relation types. We provide more experimental results for each dataset ratio with breakdown by query type in Appendix C.

### 5.4 Ranking of Easy and Hard Answers

In addition to evaluating *faithfullness* that measures whether a model could recover easy answers, it is also insightful to measure whether all easy answers can be ranked higher than hard answers. That is, a reliable query answering model would first recover all possible easy answers and would enrich the answer set with highly-probable hard answers. To this end, we apply a ROC AUC metric over original **unfiltered** scores. The ROC AUC score measures how many hard answers are ranked *after* easy answers. Note that the score does not depend on actual values of ranks, that is, the metric will be high when easy answers are ranked, e.g., in between 100-1000 as long as hard answers are ranked 1001 and lower. Therefore, ROC AUC still needs to be paired with MRR to see how easy and hard answers are ranked absolutely.

Table 2: Macro-averaged ROC AUC score over **unfiltered** predictions on the reference $\mathcal{E}_{inf}/\mathcal{E}_{train} = 175\%$ dataset to measure if all easy answers are ranked higher than hard answers. Higher is better.

| Model | avg$_p$ | avg$_n$ | 1p | 2p | 3p | 2i | 3i | pi | ip | 2u | up | 2in | 3in | inp | pin | pni |
|---|---|---|---|---|---|---|---|---|---|---|---|---|---|---|---|---|
| | | | | | | *Inductive Inference-only* | | | | | | | | | | |
| NodePiece-QE | 0.692 | | 0.623 | 0.710 | 0.711 | 0.657 | 0.654 | 0.692 | 0.731 | 0.723 | 0.729 | | | | | |
| NodePiece-QE w/ GNN | 0.776 | | 0.783 | 0.783 | 0.739 | 0.758 | 0.733 | 0.760 | 0.801 | 0.841 | 0.787 | | | | | |
| | | | | | | *Inductive Trainable* | | | | | | | | | | |
| GNN-QE | 0.973 | 0.885 | 0.998 | 0.992 | 0.986 | 0.969 | 0.962 | 0.967 | 0.969 | 0.938 | 0.978 | 0.879 | 0.859 | 0.926 | 0.914 | 0.847 |

We compute ROC AUC for each query and average them over each query type thus making it **macro-averaged ROC AUC**. Our experimental results on all query types using the models reported in Table 1 on the reference 175% dataset are compiled in Table 2.

GNN-QE has nearly perfect ROC AUC scores as it was trained on complex queries. NodePiece-QE models are acceptable for inference-only models that were only trained only on 1p simple link prediction and have never seen any complex query at training time.

### 5.5 Scaling to Millions of Nodes on WikiKG-QE

Finally, we perform a scalability experiment evaluating complex query answering in the inductive mode on a new large dataset *WikiKG-QE* constructed from OGB WikiKG 2 [16] (CC0 license). While the original task is transductive link prediction, we split the graph into a training graph of 1.5M entities (5.8M edges, 512 unique relation types) and validation (test) graphs of 500k unseen nodes (5M known and 600k missing edges) each. The resulting validation (test) inference graphs are therefore of 2M entities and 11M edges with the $\mathcal{E}_{inf}/\mathcal{E}_{train}$ ratio of $133\%$ (details are in Appendix B).

GNN-QE cannot scale to such sizes, so we only probe NodePiece-QE models. Due to the problem size, we only sample 10k EPFO queries of each type from the *test inference* graph to run in the inference-only regime. Each query has at least one missing edge to be predicted at inference. The answers are ranked against all 2M entities in the filtered setting (in contrast to the OGB task that ranks against 1000 pre-computed negative samples) and Hits@100 as the target metric.

We pre-train a NodePiece encoder (in addition to relation types, we tokenize nodes with a vocabulary of 20k nodes, total 3M parameters in the encoder) with the ComplEx decoder on *1p* link prediction over the training graph for 1M steps (see Appendix D for hyperparameters). Then, the graph is extended with 500k new nodes and 5M new edges forming the inference graph. Then, using the pre-trained encoder, we materialize representations of entities (both seen and new) and relations from this inference graph. Finally, CQD-Beam executes the queries against the bigger inference graph extended with 500k new nodes and 5M new edges.

Table 3: Test Hits@100 of NodePiece-QE on WikiKG-QE (2M nodes, 11M edges including 500k new nodes and 5M new edges) in the inference-only regime. $\text{avg}_p$ is the average on EPFO queries.

| Model | $\text{avg}_p$ | 1p | 2p | 3p | 2i | 3i | pi | ip | 2u | up |
|---|---|---|---|---|---|---|---|---|---|---|
| Edge-type Heuristic | 3.1 | 10.0 | 1.0 | 0.9 | 3.7 | 8.1 | 1.8 | 0.9 | 0.7 | 0.5 |
| NodePiece-QE | 9.2 | 22.6 | 5.2 | 3.9 | 11.6 | 17.4 | 7.0 | 4.5 | 7.4 | 3.2 |
| NodePiece-QE w/ GNN | 10.1 | 66.6 | 0.9 | 0.6 | 5.4 | 8.2 | 2.3 | 0.8 | 5.2 | 0.5 |

As shown in Table 3, we find a non-trivial performance of the inference-only model on EPFO queries demonstrating that inductive *node representation* QE models are able to scale to graphs with hundreds of thousands of new nodes and millions of new edges in the zero-shot fashion. That is, answering complex queries over unseen entities is available right upon updating the graph without the need to retrain a model. This fact paves the way for the concept of *neural graph databases* capable of performing zero-shot inference over updatable graphs without expensive retraining.

## 6   Limitations and Future Work

**Limitations.** With the two proposed inductive query answering strategies, we observe a common trade-off between the performance and computational complexity. That is, inductive *node representation* models like NodePiece-QE are fast, scalable, and can be executed in the inference-only regime but underperform compared to the inductive *relational structure representation* models like GNN-QE. On the other hand, GNN-QE incurs high computational costs due to executing each query on a uniquely initialized graph instance. Alleviating this issue is a key to scalability.

**Societal Impact.** The inductive setup assumes running inference on (partly) unseen data, that is, the nature of this unseen data might be out-of-distrbution, unknown and potentially malicious. This fact has to be taken into account when evaluating predictions and overall system trustworthiness.

**Conclusion and Future Work.** In this work, we defined the problem of inductive complex logical query answering and proposed two possible parameterization strategies based on *node* and *relational structure* representations to deal with new, unseen entities at inference time. Experiments demonstrated that both strategies are able to answer complex logical queries over unseen entities as well as identify new answers on larger inference graphs. In the future work, we plan to extend the inductive setup to completely disjoint training and inference graphs, expand the set of supported logical query patterns aligned with popular queries over real-world KGs, enable reasoning over continuous features like texts and numbers, support more KG modalities like hypergraphs and hyper-relational graphs, and further explore the concept of neural graph databases.

## Acknowledgments and Disclosure of Funding

This project is supported by the Natural Sciences and Engineering Research Council (NSERC) Discovery Grant, the Canada CIFAR AI Chair Program, collaboration grants between Microsoft Research and Mila, Samsung Electronics Co., Ltd., Amazon Faculty Research Award, Tencent AI Lab Rhino-Bird Gift Fund and a NRC Collaborative R&D Project (AI4D-CORE-06). This project was also partially funded by IVADO Fundamental Research Project grant PRF-2019-3583139727. The computation resource of this project is supported by Calcul Québec[5] and Compute Canada[6]. We would like to thank anonymous reviewers for the comments that helped to improve the manuscript.

---

[5] https://www.calculquebec.ca/
[6] https://www.computecanada.ca/

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
