## A    Differentiable Logical Operators

T-norms ($\top$) and t-conorms ($\bot$) are *fuzzy* versions of conjunction ($\land$) and disjunction ($\lor$), respectively. Fuzzy operators can be applied to vectors of continuous values within a certain range, e.g., $[0,1]^d$, depending on the chosen *fuzzy logic*, and are executed as algebraic operations which makes them differentiable. Different *fuzzy logics* implement different t-norms and t-conorms. In this work, we experiment with two such logics: *product logic* and *Gödel (min) logic*. In the product logic, conjunction $\mathcal{C}$, disjunction $\mathcal{D}$, and negation $\mathcal{N}$ are modeled as follows:

$$\mathcal{C}(\boldsymbol{x}, \boldsymbol{y}) = \boldsymbol{x} \odot \boldsymbol{y}$$
$$\mathcal{D}(\boldsymbol{x}, \boldsymbol{y}) = \boldsymbol{x} + \boldsymbol{y} - \boldsymbol{x} \odot \boldsymbol{y}$$
$$\mathcal{N}(\boldsymbol{x}) = \boldsymbol{1} - \boldsymbol{x}$$

where inputs $\boldsymbol{x}, \boldsymbol{y} \in [0,1]^d$ are $d$-dimensional vectors with values in the range $[0,1]$, $\odot$ is the element-wise multiplication, and $\boldsymbol{1}$ is the *universe* vector of all ones.

In the Gödel logic, conjunction $\mathcal{C}$ and disjunction $\mathcal{D}$ are modeled as *min* and *max*, respectively:

$$\mathcal{C}(\boldsymbol{x}, \boldsymbol{y}) = min(\boldsymbol{x}, \boldsymbol{y})$$
$$\mathcal{D}(\boldsymbol{x}, \boldsymbol{y}) = max(\boldsymbol{x}, \boldsymbol{y})$$

For GNN-QE we employ solely the product logic for end-to-end training on all types of complex queries. For NodePiece-QE and its inference-only mechanism based on CQD-Beam, we may select the best performing logic for each query type based on the validation set. The chosen operators for NodePiece-QE are reported in Table 13 in Appedix D.

## B    Benchmarking Datasets Details

We sampled 9 datasets (used in Section 5.2 and Section 5.3) from the original FB15k-237 [29] with already added inverse edges for ensuring reachability and connectedness of the underlying graph for the subsequent query sampling. Creation details are provided in the Section 5.1 and statistics on the sampled graphs are presented in Table 4. Varying the ratio of entities in the inference graph to the training graph $\mathcal{E}_{inf}/\mathcal{E}_{train}$, we aim at measuring inductive capabilities of proposed strategies in the out-of-distribution size generalization scenario. To measure scalability of inductive query answering approaches, we create WikiKG-QE, an inductive split of the originally transductive OGB WikiKG 2 [16], following the same sampling strategy as for 9 Freebase datasets.

Table 4: Sampled graphs statistics for various ratios $\mathcal{E}_{inf}/\mathcal{E}_{train}$. Originally inverse triples are included in all graphs except WikiKG-QE. $\mathcal{R}$ - number of unique relation types, $\mathcal{E}$ - number of entities in various splits, $\mathcal{T}$ - number of triples. Validation and Test splits contain an inference graph $(\mathcal{E}_{inf}, \mathcal{T}_{inf})$ which is a superset of the training graph with new nodes, and missing edges to predict $\mathcal{T}_{pred}$.

| Ratio, % | $\mathcal{R}$ | $\mathcal{E}_{total}$ | Training Graph | | Validation Graph | | | Test Graph | | |
| --- | --- | --- | --- | --- | --- | --- | --- | --- | --- | --- |
| | | | $\mathcal{E}_{train}$ | $\mathcal{T}_{train}$ | $\mathcal{E}_{inf}^{val}$ | $\mathcal{T}_{inf}^{val}$ | $\mathcal{T}_{pred}^{val}$ | $\mathcal{E}_{inf}^{test}$ | $\mathcal{T}_{inf}^{test}$ | $\mathcal{T}_{pred}^{test}$ |
| 106% | 466 | 14,512 | 13,091 | 493,425 | 13,801 | 551,336 | 10,219 | 13,802 | 538,896 | 8,023 |
| 113% | 468 | 14,442 | 11,601 | 401,677 | 13,022 | 491,518 | 15,849 | 13,021 | 486,068 | 14,893 |
| 122% | 466 | 14,444 | 10,184 | 298,879 | 12,314 | 413,554 | 20,231 | 12,314 | 430,892 | 23,289 |
| 134% | 466 | 14,305 | 8,634 | 228,729 | 11,468 | 373,262 | 25,477 | 11,471 | 367,810 | 24,529 |
| 150% | 462 | 14,333 | 7,232 | 162,683 | 10,783 | 311,462 | 26,235 | 10,782 | 331,352 | 29,755 |
| 175% | 436 | 14,022 | 5,560 | 102,521 | 9,801 | 265,412 | 28,691 | 9,781 | 266,494 | 28,891 |
| 217% | 446 | 13,986 | 4,134 | 52,455 | 9,062 | 227,284 | 30,809 | 9,058 | 212,386 | 28,177 |
| 300% | 412 | 13,868 | 2,650 | 24,439 | 8,252 | 178,680 | 27,135 | 8,266 | 187,156 | 28,657 |
| 550% | 312 | 13,438 | 1,084 | 5,265 | 7,247 | 136,558 | 22,981 | 7,275 | 133,524 | 22,503 |
| | | | | | *WikiKG-QE* | | | | | |
| 133% | 512 | 2,492,122 | 1,494,033 | 5,824,868 | 1,992,739 | 9,466,319 | 638,389 | 1,993,416 | 10,510,906 | 824,713 |

In all datasets, entities and relations are anonymized and only have an integer ID. Furthermode, inference graphs at validation and test time are supersets of the respective training graph with new nodes and edges. The amount of new unique nodes is simply the difference $\mathcal{E}_{inf} - \mathcal{E}_{train}$ between entities in those graphs, e.g., for the dataset of ratio $175\%$, the validation inference graph contains $4,241$ new nodes and test inference graph contains $4,221$ news nodes. Note that those $4,241$ and $4,221$ nodes are unique for each graph and do not overlap. That is, validation inference and test inference graphs are disconnected except sharing the same core training graph.

Then, for each created inductive dataset, we sample queries of 14 query patterns following the BetaE [24] procedure. That is, *training* queries are sampled from the *training* graph $\mathcal{G}_{train}$ and have only *easy* answers reachable by simple edge traversal. Validation and test queries are sampled from the respective splits, e.g., *validation* queries are sampled from the validation graph $\mathcal{G}_{val}$ using entities from the validation inference graph $\mathcal{E}_{inf}^{val}$ (which, in turn, are a union of training nodes and new, unseen validation nodes $\mathcal{E}_{train} \cup \mathcal{E}_{val}$), and at least one edge in each query belongs to $\mathcal{T}_{pred}^{val}$ and has to be predicted during query execution. Queries might have *easy* answers that are directly reachable by traversing edges $\mathcal{T}_{inf}^{val}$ in the validation inference graph, whereas *hard* answers are only reachable after predicting missing edges from the set $\mathcal{T}_{pred}^{val}$. Final evaluation metrics are computed only based on the *hard* answers. Following the literature [24], we only retain queries that have less than 1000 answers. Table 5 summarizes the statistics on the sampled queries for each dataset ratio, each graph, and query type that we use in Section 5.2 for evaluating inductive query answering performance. In graphs with smaller inference graphs and smaller number of missing triples, we sample fewer queries with negation (*2in, 3in, inp, pin, pni*) for validation and test splits. For WikiKG-QE, due to its size, we only sample 10k EPFO queries to be executed in the inference-only regime without training (at the moment, CQD-Beam does not support queries with negation). We use those queries in Section 5.5 to evaluate scalability of NodePiece-QE and prediction quality in the inference-only mode.

Table 5: Statistics on sampled queries for each dataset ratio and query type. For WikiKG-QE, we only sample EPFO queries without negation.

| Ratio | Graph | 1p | 2p | 3p | 2i | 3i | pi | ip | 2u | up | 2in | 3in | inp | pin | pni |
|---|---|---|---|---|---|---|---|---|---|---|---|---|---|---|---|
| 106% | training | 135,613 | 50,000 | 50,000 | 50,000 | 50,000 | 50,000 | 50,000 | 50,000 | 50,000 | 50,000 | 40,000 | 50,000 | 50,000 | 50,000 |
| | validation | 6,582 | 10,000 | 10,000 | 10,000 | 10,000 | 10,000 | 10,000 | 10,000 | 10,000 | 1,000 | 1,000 | 1,000 | 1,000 | 1,000 |
| | test | 5,446 | 10,000 | 10,000 | 10,000 | 10,000 | 10,000 | 10,000 | 10,000 | 10,000 | 1,000 | 1,000 | 1,000 | 1,000 | 1,000 |
| 113% | training | 115,523 | 50,000 | 50,000 | 50,000 | 50,000 | 50,000 | 50,000 | 50,000 | 50,000 | 50,000 | 40,000 | 50,000 | 50,000 | 50,000 |
| | validation | 10,256 | 10,000 | 10,000 | 10,000 | 10,000 | 10,000 | 10,000 | 10,000 | 10,000 | 1,000 | 1,000 | 1,000 | 1,000 | 1,000 |
| | test | 9,782 | 10,000 | 10,000 | 10,000 | 10,000 | 10,000 | 10,000 | 10,000 | 10,000 | 1,000 | 1,000 | 1,000 | 1,000 | 1,000 |
| 122% | training | 91,228 | 50,000 | 50,000 | 50,000 | 50,000 | 50,000 | 50,000 | 50,000 | 50,000 | 50,000 | 40,000 | 50,000 | 50,000 | 50,000 |
| | validation | 12,696 | 10,000 | 10,000 | 10,000 | 10,000 | 10,000 | 10,000 | 10,000 | 10,000 | 5,000 | 5,000 | 5,000 | 5,000 | 5,000 |
| | test | 14,458 | 10,000 | 10,000 | 10,000 | 10,000 | 10,000 | 10,000 | 10,000 | 10,000 | 5,000 | 5,000 | 5,000 | 5,000 | 5,000 |
| 134% | training | 75,326 | 50,000 | 50,000 | 50,000 | 50,000 | 50,000 | 50,000 | 50,000 | 50,000 | 50,000 | 40,000 | 50,000 | 50,000 | 50,000 |
| | validation | 15,541 | 50,000 | 50,000 | 50,000 | 50,000 | 50,000 | 50,000 | 20,000 | 20,000 | 5,000 | 5,000 | 5,000 | 5,000 | 5,000 |
| | test | 15,270 | 50,000 | 50,000 | 50,000 | 50,000 | 50,000 | 50,000 | 20,000 | 20,000 | 5,000 | 5,000 | 5,000 | 5,000 | 5,000 |
| 150% | training | 56,114 | 50,000 | 50,000 | 50,000 | 50,000 | 50,000 | 50,000 | 50,000 | 50,000 | 50,000 | 40,000 | 50,000 | 50,000 | 50,000 |
| | validation | 16,229 | 50,000 | 50,000 | 50,000 | 50,000 | 50,000 | 50,000 | 50,000 | 50,000 | 5,000 | 5,000 | 5,000 | 5,000 | 5,000 |
| | test | 17,683 | 50,000 | 50,000 | 50,000 | 50,000 | 50,000 | 50,000 | 50,000 | 50,000 | 5,000 | 5,000 | 5,000 | 5,000 | 5,000 |
| 175% | training | 38,851 | 50,000 | 50,000 | 50,000 | 50,000 | 50,000 | 50,000 | 50,000 | 50,000 | 50,000 | 40,000 | 50,000 | 50,000 | 50,000 |
| | validation | 17,235 | 50,000 | 50,000 | 50,000 | 50,000 | 50,000 | 50,000 | 50,000 | 50,000 | 10,000 | 10,000 | 10,000 | 10,000 | 10,000 |
| | test | 17,476 | 50,000 | 50,000 | 50,000 | 50,000 | 50,000 | 50,000 | 50,000 | 50,000 | 10,000 | 10,000 | 10,000 | 10,000 | 10,000 |
| 217% | training | 22,422 | 30,000 | 30,000 | 50,000 | 50,000 | 50,000 | 50,000 | 50,000 | 50,000 | 30,000 | 30,000 | 50,000 | 50,000 | 50,000 |
| | validation | 18,168 | 50,000 | 50,000 | 50,000 | 50,000 | 50,000 | 50,000 | 50,000 | 50,000 | 10,000 | 10,000 | 10,000 | 10,000 | 10,000 |
| | test | 16,902 | 50,000 | 50,000 | 50,000 | 50,000 | 50,000 | 50,000 | 50,000 | 50,000 | 10,000 | 10,000 | 10,000 | 10,000 | 10,000 |
| 300% | training | 11,699 | 15,000 | 15,000 | 40,000 | 40,000 | 50,000 | 50,000 | 50,000 | 50,000 | 15,000 | 15,000 | 50,000 | 40,000 | 50,000 |
| | validation | 16,189 | 50,000 | 50,000 | 50,000 | 50,000 | 50,000 | 50,000 | 50,000 | 50,000 | 10,000 | 10,000 | 10,000 | 10,000 | 10,000 |
| | test | 17,105 | 50,000 | 50,000 | 50,000 | 50,000 | 50,000 | 50,000 | 50,000 | 50,000 | 10,000 | 10,000 | 10,000 | 10,000 | 10,000 |
| 550% | training | 3,284 | 15,000 | 15,000 | 40,000 | 40,000 | 50,000 | 50,000 | 50,000 | 50,000 | 10,000 | 10,000 | 30,000 | 30,000 | 30,000 |
| | validation | 13,616 | 50,000 | 50,000 | 50,000 | 50,000 | 50,000 | 50,000 | 50,000 | 50,000 | 10,000 | 10,000 | 10,000 | 10,000 | 10,000 |
| | test | 13,670 | 50,000 | 50,000 | 50,000 | 50,000 | 50,000 | 50,000 | 50,000 | 50,000 | 10,000 | 10,000 | 10,000 | 10,000 | 10,000 |
| | | | | | | | *WikiKG-QE* | | | | | | | | | |
| 133% | training | 10,000 | 10,000 | 10,000 | 10,000 | 10,000 | 10,000 | 10,000 | 10,000 | 10,000 | - | - | - | - | - |
| | validation | 10,000 | 10,000 | 10,000 | 10,000 | 10,000 | 10,000 | 10,000 | 10,000 | 10,000 | - | - | - | - | - |
| | test | 10,000 | 10,000 | 10,000 | 10,000 | 10,000 | 10,000 | 10,000 | 10,000 | 10,000 | - | - | - | - | - |

Furthermore, for the experiment in Section 5.3 to measure the abilities of inductive models to find new answers of known queries, we take the created **training** queries and find their *easy* answers in the validation inference $\mathcal{G}_{inf}^{val} = (\mathcal{E}_{inf}^{val}, \mathcal{T}_{inf}^{val})$ and test inference $\mathcal{G}_{inf}^{test} = (\mathcal{E}_{inf}^{test}, \mathcal{T}_{inf}^{test})$ graphs. That is, those new answers do not require predicting missing edges in the inference graphs and only require a model to execute edge traversal to find (if any) new correct answers involving new, unseen entities and edges. For the validation (test) split, we only count such training queries $q$ whose answer set in

Table 6: Statistics on **training** EPFO queries that have a different (often, larger) answer set when executed against validation and test inference graphs. We list the original number of training queries, number of those queries with new *easy* answers in the validation (In val) and test graphs (In test), as well as their percentage ratio to the total number. Most queries (except *2i,3i*) have new answer sets.

| Ratio | Graph | 1p #Q | 1p % | 2p #Q | 2p % | 3p #Q | 3p % | 2i #Q | 2i % | 3i #Q | 3i % | pi #Q | pi % | ip #Q | ip % | 2u #Q | 2u % | up #Q | up % |
|---|---|---|---|---|---|---|---|---|---|---|---|---|---|---|---|---|---|---|---|
| | Train | 135,613 | 100.0 | 50,000 | 100.0 | 50,000 | 100.0 | 50,000 | 100.0 | 50,000 | 100.0 | 50,000 | 100.0 | 50,000 | 100.0 | 50,000 | 100.0 | 50,000 | 100.0 |
| 106% | In val | 14,079 | 10.4 | 32,220 | 64.4 | 40,860 | 81.7 | 7,598 | 15.2 | 4,416 | 8.8 | 16,485 | 33.0 | 29,290 | 58.6 | 33,507 | 67.0 | 41,671 | 83.3 |
| | In test | 11,560 | 8.5 | 31,894 | 63.8 | 40,547 | 81.1 | 7,313 | 14.6 | 4,175 | 8.4 | 16,204 | 32.4 | 28,778 | 57.6 | 32,978 | 66.0 | 41,167 | 82.3 |
| | Train | 115,523 | 100.0 | 50,000 | 100.0 | 50,000 | 100.0 | 50,000 | 100.0 | 50,000 | 100.0 | 50,000 | 100.0 | 50,000 | 100.0 | 50,000 | 100.0 | 50,000 | 100.0 |
| 113% | In val | 17,792 | 15.4 | 36,499 | 73.0 | 43,473 | 86.9 | 10,517 | 21.0 | 6,394 | 12.8 | 20,556 | 41.1 | 33,599 | 67.2 | 37,955 | 75.9 | 44,318 | 88.6 |
| | In test | 17,576 | 15.2 | 36,721 | 73.4 | 43,541 | 87.1 | 10,552 | 21.1 | 6,303 | 12.6 | 20,382 | 40.8 | 33,726 | 67.5 | 38,107 | 76.2 | 44,501 | 89.0 |
| | Train | 91,228 | 100.0 | 50,000 | 100.0 | 50,000 | 100.0 | 50,000 | 100.0 | 50,000 | 100.0 | 50,000 | 100.0 | 50,000 | 100.0 | 50,000 | 100.0 | 50,000 | 100.0 |
| 122% | In val | 20,281 | 22.2 | 38,642 | 77.3 | 44,654 | 89.3 | 11,695 | 23.4 | 5,851 | 11.7 | 22,662 | 45.3 | 35,935 | 71.9 | 40,356 | 80.7 | 45,672 | 91.3 |
| | In test | 20,418 | 22.4 | 38,706 | 77.4 | 44,688 | 89.4 | 11,847 | 23.7 | 6,185 | 12.4 | 22,524 | 45.0 | 35,768 | 71.5 | 40,395 | 80.8 | 45,684 | 91.4 |
| | Train | 75,326 | 100.0 | 50,000 | 100.0 | 50,000 | 100.0 | 50,000 | 100.0 | 50,000 | 100.0 | 50,000 | 100.0 | 50,000 | 100.0 | 50,000 | 100.0 | 50,000 | 100.0 |
| 134% | In val | 18,909 | 25.1 | 39,893 | 79.8 | 45,253 | 90.5 | 14,256 | 28.5 | 8,655 | 17.3 | 24,619 | 49.2 | 37,835 | 75.7 | 41,899 | 83.8 | 46,114 | 92.2 |
| | In test | 19,372 | 25.7 | 39,762 | 79.5 | 45,325 | 90.7 | 14,082 | 28.2 | 8,790 | 17.6 | 24,212 | 48.4 | 37,527 | 75.1 | 41,494 | 83.0 | 46,210 | 92.4 |
| | Train | 56,114 | 100.0 | 50,000 | 100.0 | 50,000 | 100.0 | 50,000 | 100.0 | 50,000 | 100.0 | 50,000 | 100.0 | 50,000 | 100.0 | 50,000 | 100.0 | 50,000 | 100.0 |
| 150% | In val | 17,434 | 31.1 | 40,666 | 81.3 | 45,832 | 91.7 | 14,103 | 28.2 | 8,011 | 16.0 | 25,106 | 50.2 | 38,499 | 77.0 | 42,587 | 85.2 | 46,754 | 93.5 |
| | In test | 18,566 | 33.1 | 41,202 | 82.4 | 46,092 | 92.2 | 14,575 | 29.2 | 8,193 | 16.4 | 25,782 | 51.6 | 38,932 | 77.9 | 43,101 | 86.2 | 46,791 | 93.6 |
| | Train | 38,851 | 100.0 | 50,000 | 100.0 | 50,000 | 100.0 | 50,000 | 100.0 | 50,000 | 100.0 | 50,000 | 100.0 | 50,000 | 100.0 | 50,000 | 100.0 | 50,000 | 100.0 |
| 175% | In val | 14,063 | 36.2 | 41,290 | 82.6 | 46,214 | 92.4 | 15,645 | 31.3 | 9,222 | 18.4 | 27,205 | 54.4 | 40,161 | 80.3 | 44,128 | 88.3 | 47,366 | 94.7 |
| | In test | 14,214 | 36.6 | 41,143 | 82.3 | 46,061 | 92.1 | 15,731 | 31.5 | 9,391 | 18.8 | 27,207 | 54.4 | 40,297 | 80.6 | 43,980 | 88.0 | 47,319 | 94.6 |
| | Train | 22,422 | 100.0 | 30,000 | 100.0 | 30,000 | 100.0 | 50,000 | 100.0 | 50,000 | 100.0 | 50,000 | 100.0 | 50,000 | 100.0 | 50,000 | 100.0 | 50,000 | 100.0 |
| 217% | In val | 10,437 | 46.5 | 24,659 | 82.2 | 26,760 | 89.2 | 13,784 | 27.6 | 7,807 | 15.6 | 24,884 | 49.8 | 39,107 | 78.2 | 43,496 | 87.0 | 46,112 | 92.2 |
| | In test | 10,257 | 45.7 | 24,344 | 81.1 | 26,579 | 88.6 | 14,055 | 28.1 | 7,962 | 15.9 | 24,962 | 49.9 | 38,966 | 77.9 | 43,092 | 86.2 | 45,850 | 91.7 |
| | Train | 11,699 | 100.0 | 15,000 | 100.0 | 15,000 | 100.0 | 40,000 | 100.0 | 40,000 | 100.0 | 50,000 | 100.0 | 50,000 | 100.0 | 50,000 | 100.0 | 50,000 | 100.0 |
| 300% | In val | 5,830 | 49.8 | 12,366 | 82.4 | 13,230 | 88.2 | 12,833 | 32.1 | 7,911 | 19.8 | 27,920 | 55.8 | 40,800 | 81.6 | 43,516 | 87.0 | 46,453 | 92.9 |
| | In test | 6,061 | 51.8 | 12,477 | 83.2 | 13,309 | 88.7 | 13,291 | 33.2 | 8,284 | 20.7 | 28,447 | 56.9 | 41,214 | 82.4 | 43,966 | 87.9 | 46,668 | 93.3 |
| | Train | 3,284 | 100.0 | 15,000 | 100.0 | 15,000 | 100.0 | 40,000 | 100.0 | 40,000 | 100.0 | 50,000 | 100.0 | 50,000 | 100.0 | 50,000 | 100.0 | 50,000 | 100.0 |
| 550% | In val | 1,885 | 57.4 | 11,484 | 76.6 | 12,575 | 83.8 | 11,119 | 27.8 | 6,617 | 16.5 | 23,126 | 46.3 | 39,243 | 78.5 | 38,129 | 76.3 | 45,173 | 90.3 |
| | In test | 1,883 | 57.3 | 11,597 | 77.3 | 12,654 | 84.4 | 11,244 | 28.1 | 6,795 | 17.0 | 23,575 | 47.2 | 39,630 | 79.3 | 37,508 | 75.0 | 45,412 | 90.8 |

Table 7: Statistics on **training** negation queries that have a different (often, larger) answer set when executed against validation and test inference graphs. We list the original number of training queries, number of those queries with new *easy* answers in the validation (In val) and test graphs (In test), as well as their percentage ratio to the total number. Most queries have new answer sets.

| Ratio | Graph | 2in #Q | 2in % | 3in #Q | 3in % | pin #Q | pin % | pni #Q | pni % | inp #Q | inp % |
|---|---|---|---|---|---|---|---|---|---|---|---|
| | Train | 50,000 | 100.0 | 50,000 | 100.0 | 50,000 | 100.0 | 50,000 | 100.0 | 50,000 | 100.0 |
| 106% | In val | 25,318 | 50.6 | 18,232 | 36.5 | 37,857 | 75.7 | 27,572 | 55.1 | 37,497 | 75.0 |
| | In test | 25,111 | 50.2 | 18,237 | 36.5 | 37,441 | 74.9 | 27,535 | 55.1 | 37,176 | 74.4 |
| | Train | 50,000 | 100.0 | 50,000 | 100.0 | 50,000 | 100.0 | 50,000 | 100.0 | 50,000 | 100.0 |
| 113% | In val | 31,216 | 62.4 | 24,620 | 49.2 | 42,015 | 84.0 | 33,011 | 66.0 | 41,980 | 84.0 |
| | In test | 31,437 | 62.9 | 24,665 | 49.3 | 42,255 | 84.5 | 33,115 | 66.2 | 42,296 | 84.6 |
| | Train | 50,000 | 100.0 | 50,000 | 100.0 | 50,000 | 100.0 | 50,000 | 100.0 | 50,000 | 100.0 |
| 122% | In val | 34,722 | 69.4 | 26,700 | 53.4 | 44,104 | 88.2 | 36,361 | 72.7 | 44,070 | 88.1 |
| | In test | 35,028 | 70.1 | 27,105 | 54.2 | 44,089 | 88.2 | 36,398 | 72.8 | 44,074 | 88.1 |
| | Train | 50,000 | 100.0 | 50,000 | 100.0 | 50,000 | 100.0 | 50,000 | 100.0 | 50,000 | 100.0 |
| 134% | In val | 38,096 | 76.2 | 31,631 | 63.3 | 45,672 | 91.3 | 39,641 | 79.3 | 45,491 | 91.0 |
| | In test | 37,469 | 74.9 | 31,224 | 62.4 | 45,521 | 91.0 | 38,971 | 77.9 | 45,418 | 90.8 |
| | Train | 50,000 | 100.0 | 40,000 | 100.0 | 50,000 | 100.0 | 50,000 | 100.0 | 50,000 | 100.0 |
| 150% | In val | 39,836 | 79.7 | 26,534 | 66.3 | 46,561 | 93.1 | 40,733 | 81.5 | 46,496 | 93.0 |
| | In test | 40,127 | 80.3 | 26,968 | 67.4 | 46,832 | 93.7 | 41,100 | 82.2 | 46,811 | 93.6 |
| | Train | 50,000 | 100.0 | 40,000 | 100.0 | 50,000 | 100.0 | 50,000 | 100.0 | 50,000 | 100.0 |
| 175% | In val | 42,418 | 84.8 | 29,083 | 72.7 | 47,666 | 95.3 | 42,987 | 86.0 | 47,606 | 95.2 |
| | In test | 42,379 | 84.8 | 29,170 | 72.9 | 47,749 | 95.5 | 42,941 | 85.9 | 47,557 | 95.1 |
| | Train | 30,000 | 100.0 | 30,000 | 100.0 | 50,000 | 100.0 | 50,000 | 100.0 | 50,000 | 100.0 |
| 217% | In val | 26,202 | 87.3 | 21,751 | 72.5 | 47,879 | 95.8 | 43,958 | 87.9 | 47,688 | 95.4 |
| | In test | 26,080 | 86.9 | 21,591 | 72.0 | 47,655 | 95.3 | 43,837 | 87.7 | 47,417 | 94.8 |
| | Train | 15,000 | 100.0 | 15,000 | 100.0 | 50,000 | 100.0 | 40,000 | 100.0 | 50,000 | 100.0 |
| 300% | In val | 13,595 | 90.6 | 11,996 | 80.0 | 48,693 | 97.4 | 36,427 | 91.1 | 48,279 | 96.6 |
| | In test | 13,659 | 91.1 | 12,098 | 80.7 | 48,791 | 97.6 | 36,507 | 91.3 | 48,440 | 96.9 |
| | Train | 10,000 | 100.0 | 10,000 | 100.0 | 30,000 | 100.0 | 30,000 | 100.0 | 30,000 | 100.0 |
| 550% | In val | 9,232 | 92.3 | 8,071 | 80.7 | 29,484 | 98.3 | 27,975 | 93.3 | 29,393 | 98.0 |
| | In test | 9,137 | 91.4 | 8,053 | 80.5 | 29,510 | 98.4 | 27,839 | 92.8 | 29,218 | 97.4 |

this split is *different* from the answer set in the training graph, e.g., $\mathcal{A}_q^{val} \neq \mathcal{A}_q^{train}$. We summarize the statistics of identified new answer sets in all datasets in Table 6 (for EPFO queries) and Table 7 (for queries with negations). We find that in most query patterns across all dataset ratios, training queries indeed have new answer sets when executed against validation or test inference graphs.

## C   More Experimental Results

Here, we present a detailed breakdown of query answering performance measured in Sections 5.2 and 5.3 by query type. Fig. 5 and Table 8 contain detailed results from Section 5.2 of executing **test** queries with new, unseen entities over inference graphs of various ratios of new entities.

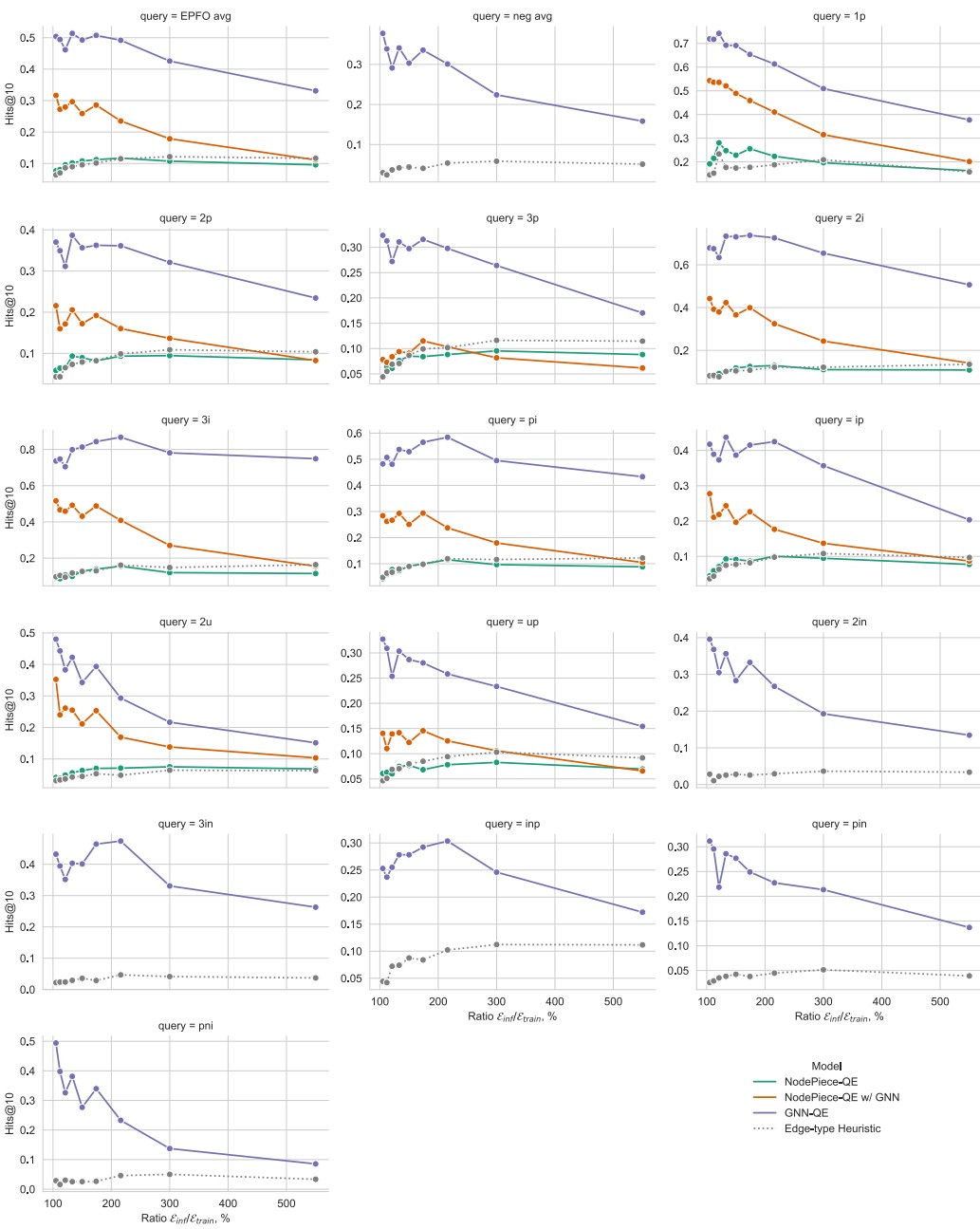

Figure 5: Hits@10 results on answering **test** inductive FOL queries on all ratios $\mathcal{E}_{inf}/\mathcal{E}_{train}$.

Table 8: Test Hits@3 and Hits@10 results (%) on answering **test** inductive FOL queries on all ratios $\mathcal{E}_{inf}/\mathcal{E}_{train}$. $\text{avg}_p$ is the average on EPFO queries ($\wedge$, $\vee$). $\text{avg}_n$ is the average on queries with negation.

| Ratio | Model | Metric | avg_p | avg_n | 1p | 2p | 3p | 2i | 3i | pi | ip | 2u | up | 2in | 3in | inp | pin | pni |
|---|---|---|---|---|---|---|---|---|---|---|---|---|---|---|---|---|---|---|
| 550% | Edge-type Heuristic | Hits@3 | 5.0 | 2.3 | 5.9 | 4.7 | 5.2 | 5.3 | 7.0 | 5.1 | 4.3 | 2.8 | 4.5 | 1.5 | 1.7 | 4.9 | 2.0 | 1.4 |
| | | Hits@10 | 11.7 | 5.1 | 15.8 | 10.4 | 11.5 | 13.4 | 16.4 | 12.2 | 9.7 | 6.3 | 9.2 | 3.4 | 3.7 | 11.2 | 3.9 | 3.4 |
| | NodePiece-QE | Hits@3 | 4.3 | | 7.3 | 4.0 | 4.1 | 4.3 | 4.5 | 3.8 | 3.6 | 3.4 | 3.3 | | | | | |
| | | Hits@10 | 9.6 | | 16.3 | 8.4 | 8.8 | 10.8 | 11.5 | 8.8 | 7.7 | 6.8 | 7.0 | | | | | |
| | NodePiece-QE w/ GNN | Hits@3 | 5.4 | | 9.2 | 4.1 | 3.1 | 6.8 | 7.4 | 5.1 | 4.5 | 5.4 | 3.4 | | | | | |
| | | Hits@10 | 11.1 | | 20.1 | 8.3 | 6.2 | 14.0 | 15.5 | 10.5 | 8.7 | 10.3 | 6.6 | | | | | |
| | GNN-QE | Hits@3 | 24.2 | 9.7 | 28.3 | 15.8 | 10.1 | 37.7 | 60.9 | 31.3 | 14.4 | 10.1 | 8.8 | 9.0 | 16.5 | 9.8 | 7.7 | 5.6 |
| | | Hits@10 | 33.1 | 15.8 | 37.7 | 23.4 | 17.0 | 50.7 | 74.9 | 43.3 | 20.4 | 15.1 | 15.4 | 13.4 | 26.3 | 17.2 | 13.7 | 8.6 |
| 300% | Edge-type Heuristic | Hits@3 | 5.5 | 2.7 | 10.3 | 5.1 | 5.4 | 5.0 | 6.3 | 5.0 | 4.8 | 2.7 | 5.1 | 1.6 | 1.9 | 4.8 | 2.5 | 2.5 |
| | | Hits@10 | 12.2 | 5.8 | 20.9 | 10.9 | 11.6 | 12.1 | 14.8 | 11.6 | 10.8 | 6.4 | 10.3 | 3.6 | 4.1 | 11.2 | 5.1 | 5.0 |
| | NodePiece-QE | Hits@3 | 5.4 | | 12.0 | 4.7 | 4.6 | 4.9 | 5.2 | 4.2 | 4.6 | 3.9 | 4.0 | | | | | |
| | | Hits@10 | 10.7 | | 19.6 | 9.5 | 9.5 | 11.0 | 12.0 | 9.7 | 9.5 | 7.5 | 8.3 | | | | | |
| | NodePiece-QE w/ GNN | Hits@3 | 9.7 | | 18.9 | 7.3 | 4.1 | 13.1 | 15.0 | 9.1 | 7.3 | 7.2 | 5.5 | | | | | |
| | | Hits@10 | 17.9 | | 31.5 | 13.7 | 8.2 | 24.3 | 27.0 | 18.0 | 13.7 | 13.8 | 10.6 | | | | | |
| | GNN-QE | Hits@3 | 31.8 | 13.5 | 41.5 | 21.0 | 16.1 | 51.7 | 66.7 | 37.2 | 25.2 | 13.5 | 13.3 | 11.7 | 21.3 | 14.0 | 12.7 | 7.7 |
| | | Hits@10 | 42.6 | 22.4 | 50.9 | 32.1 | 26.4 | 65.4 | 78.1 | 49.5 | 35.7 | 21.7 | 23.4 | 19.3 | 33.1 | 24.6 | 21.3 | 13.7 |
| 217% | Edge-type Heuristic | Hits@3 | 5.5 | 2.4 | 10.3 | 4.8 | 4.8 | 5.1 | 7.2 | 5.4 | 4.7 | 2.1 | 4.6 | 1.2 | 2.3 | 4.3 | 2.2 | 2.2 |
| | | Hits@10 | 11.5 | 5.4 | 18.8 | 9.9 | 10.2 | 12.1 | 16.1 | 11.9 | 9.8 | 4.8 | 9.4 | 2.9 | 4.7 | 10.2 | 4.5 | 4.6 |
| | NodePiece-QE | Hits@3 | 5.9 | | 13.9 | 4.8 | 4.4 | 5.8 | 6.7 | 5.3 | 5.1 | 3.4 | 4.0 | | | | | |
| | | Hits@10 | 11.7 | | 22.3 | 9.3 | 8.8 | 12.9 | 15.5 | 11.5 | 10.0 | 7.1 | 7.8 | | | | | |
| | NodePiece-QE w/ GNN | Hits@3 | 13.6 | | 25.7 | 8.8 | 5.3 | 18.7 | 24.8 | 12.9 | 10.3 | 8.9 | 6.9 | | | | | |
| | | Hits@10 | 23.5 | | 41.0 | 16.0 | 10.4 | 32.5 | 40.8 | 23.7 | 17.7 | 16.9 | 12.6 | | | | | |
| | GNN-QE | Hits@3 | 37.9 | 19.2 | 50.6 | 24.4 | 19.3 | 58.6 | 76.2 | 45.1 | 31.4 | 19.7 | 16.0 | 17.6 | 32.6 | 18.3 | 14.0 | 13.6 |
| | | Hits@10 | 49.2 | 30.1 | 61.3 | 36.1 | 29.8 | 72.6 | 86.6 | 58.4 | 42.5 | 29.3 | 25.8 | 26.7 | 47.4 | 30.3 | 22.7 | 23.3 |
| 175% | Edge-type Heuristic | Hits@3 | 4.7 | 1.7 | 8.4 | 3.8 | 4.8 | 4.5 | 5.6 | 4.3 | 4.0 | 2.5 | 4.2 | 1.0 | 1.2 | 3.3 | 1.8 | 1.0 |
| | | Hits@10 | 10.1 | 4.1 | 17.7 | 8.2 | 9.9 | 10.7 | 13.0 | 9.8 | 8.2 | 5.3 | 8.5 | 2.6 | 2.9 | 8.4 | 3.8 | 2.7 |
| | NodePiece-QE | Hits@3 | 5.6 | | 14.2 | 4.1 | 4.1 | 5.6 | 6.2 | 4.5 | 4.6 | 3.5 | 3.3 | | | | | |
| | | Hits@10 | 11.2 | | 25.5 | 8.2 | 8.4 | 12.4 | 13.9 | 9.9 | 8.7 | 7.0 | 6.8 | | | | | |
| | NodePiece-QE w/ GNN | Hits@3 | 17.2 | | 30.7 | 10.7 | 5.9 | 24.4 | 31.2 | 17.2 | 13.1 | 14.2 | 7.3 | | | | | |
| | | Hits@10 | 28.6 | | 45.9 | 19.2 | 11.5 | 39.9 | 48.8 | 29.4 | 22.6 | 25.3 | 14.6 | | | | | |
| | GNN-QE | Hits@3 | 38.5 | 20.5 | 52.8 | 24.1 | 20.6 | 59.8 | 73.3 | 43.2 | 30.0 | 24.4 | 17.9 | 18.9 | 32.2 | 17.8 | 15.3 | 18.2 |
| | | Hits@10 | 50.7 | 33.6 | 65.4 | 36.3 | 31.6 | 73.8 | 84.3 | 56.5 | 41.5 | 39.3 | 28.0 | 33.3 | 46.4 | 29.2 | 24.9 | 34.0 |
| 150% | Edge-type Heuristic | Hits@3 | 4.4 | 1.9 | 9.2 | 3.6 | 4.0 | 4.3 | 5.3 | 3.9 | 3.5 | 1.8 | 3.8 | 1.3 | 1.5 | 3.5 | 2.1 | 1.1 |
| | | Hits@10 | 9.6 | 4.4 | 17.4 | 7.9 | 8.7 | 10.4 | 12.7 | 9.0 | 7.7 | 4.5 | 8.0 | 2.8 | 3.6 | 8.7 | 4.2 | 2.5 |
| | NodePiece-QE | Hits@3 | 5.4 | | 14.0 | 4.5 | 4.1 | 5.0 | 5.5 | 4.0 | 4.6 | 3.0 | 3.8 | | | | | |
| | | Hits@10 | 10.8 | | 22.8 | 8.9 | 8.5 | 11.7 | 12.9 | 9.1 | 9.1 | 6.3 | 7.7 | | | | | |
| | NodePiece-QE w/ GNN | Hits@3 | 15.7 | | 33.1 | 9.7 | 4.6 | 22.3 | 26.9 | 14.8 | 11.4 | 12.3 | 6.5 | | | | | |
| | | Hits@10 | 25.9 | | 48.9 | 17.2 | 9.1 | 36.6 | 43.1 | 25.1 | 19.7 | 21.1 | 12.2 | | | | | |
| | GNN-QE | Hits@3 | 37.3 | 18.1 | 56.6 | 23.6 | 18.9 | 58.6 | 69.8 | 39.6 | 27.3 | 23.2 | 18.0 | 16.9 | 25.7 | 16.6 | 16.2 | 15.4 |
| | | Hits@10 | 49.3 | 30.3 | 69.1 | 35.7 | 29.7 | 73.1 | 81.3 | 52.9 | 38.7 | 34.3 | 28.7 | 28.3 | 40.1 | 27.8 | 27.7 | 27.7 |
| 133% | Edge-type Heuristic | Hits@3 | 4.0 | 1.9 | 8.6 | 3.5 | 3.2 | 4.3 | 4.9 | 3.4 | 3.5 | 1.8 | 3.2 | 1.2 | 1.6 | 2.7 | 2.0 | 1.1 |
| | | Hits@10 | 9.0 | 4.2 | 17.7 | 7.3 | 7.1 | 10.1 | 11.8 | 8.0 | 7.5 | 4.3 | 7.0 | 2.6 | 2.9 | 7.4 | 3.8 | 2.5 |
| | NodePiece-QE | Hits@3 | 5.1 | | 15.4 | 4.8 | 3.5 | 4.4 | 4.1 | 2.9 | 4.8 | 2.6 | 3.4 | | | | | |
| | | Hits@10 | 10.2 | | 24.8 | 9.3 | 7.7 | 10.1 | 9.9 | 7.4 | 9.3 | 5.6 | 7.5 | | | | | |
| | NodePiece-QE w/ GNN | Hits@3 | 19.4 | | 38.0 | 12.6 | 5.2 | 27.0 | 32.3 | 17.9 | 16.0 | 16.7 | 8.7 | | | | | |
| | | Hits@10 | 29.6 | | 52.1 | 20.6 | 9.4 | 42.3 | 49.2 | 29.3 | 24.3 | 25.5 | 14.2 | | | | | |
| | GNN-QE | Hits@3 | 38.8 | 21.4 | 56.3 | 25.6 | 19.8 | 59.3 | 68.5 | 40.6 | 30.6 | 28.4 | 19.8 | 23.0 | 25.9 | 16.4 | 18.3 | 23.6 |
| | | Hits@10 | 51.4 | 34.1 | 69.2 | 38.7 | 31.1 | 73.4 | 79.9 | 53.8 | 43.7 | 42.2 | 30.4 | 35.6 | 40.3 | 27.8 | 28.6 | 38.1 |
| 121% | Edge-type Heuristic | Hits@3 | 4.3 | 1.5 | 14.7 | 3.0 | 3.2 | 3.0 | 3.9 | 2.8 | 2.8 | 1.5 | 3.3 | 0.9 | 1.0 | 2.6 | 1.7 | 1.2 |
| | | Hits@10 | 8.6 | 3.7 | 23.3 | 6.5 | 6.9 | 7.6 | 9.5 | 6.7 | 6.4 | 3.7 | 6.9 | 2.2 | 2.4 | 7.2 | 3.5 | 3.0 |
| | NodePiece-QE | Hits@3 | 4.6 | | 16.0 | 3.2 | 2.7 | 3.7 | 4.3 | 3.1 | 3.5 | 2.1 | 2.8 | | | | | |
| | | Hits@10 | 9.6 | | 28.0 | 6.5 | 6.1 | 9.2 | 10.7 | 7.8 | 7.1 | 4.9 | 6.0 | | | | | |
| | NodePiece-QE w/ GNN | Hits@3 | 18.4 | | 39.7 | 10.6 | 4.8 | 24.8 | 30.6 | 16.4 | 16.8 | 16.8 | 7.8 | | | | | |
| | | Hits@10 | 27.9 | | 53.6 | 17.1 | 8.4 | 38.0 | 45.9 | 26.7 | 21.9 | 26.1 | 13.9 | | | | | |
| | GNN-QE | Hits@3 | 35.3 | 18.9 | 62.0 | 21.1 | 17.9 | 50.0 | 59.5 | 36.8 | 26.7 | 27.4 | 16.6 | 20.2 | 23.4 | 15.4 | 13.8 | 21.5 |
| | | Hits@10 | 46.2 | 29.1 | 74.2 | 31.1 | 27.2 | 63.4 | 70.5 | 48.1 | 37.3 | 38.3 | 25.4 | 30.5 | 35.2 | 25.5 | 21.8 | 32.6 |
| 113% | Edge-type Heuristic | Hits@3 | 3.1 | 1.0 | 8.5 | 1.8 | 2.3 | 3.3 | 4.5 | 2.5 | 1.9 | 1.3 | 2.1 | 0.5 | 1.3 | 1.5 | 1.2 | 0.4 |
| | | Hits@10 | 7.0 | 2.4 | 15.2 | 4.3 | 5.5 | 8.2 | 10.4 | 6.4 | 4.4 | 3.4 | 5.1 | 1.1 | 2.4 | 4.2 | 2.9 | 1.6 |
| | NodePiece-QE | Hits@3 | 4.0 | | 13.6 | 3.2 | 3.2 | 3.4 | 3.8 | 2.4 | 2.6 | 1.7 | 2.7 | | | | | |
| | | Hits@10 | 8.1 | | 21.5 | 6.4 | 6.4 | 7.8 | 8.6 | 5.9 | 5.9 | 3.8 | 6.3 | | | | | |
| | NodePiece-QE w/ GNN | Hits@3 | 18.1 | | 39.1 | 10.2 | 3.8 | 25.9 | 31.0 | 17.2 | 13.8 | 15.8 | 6.4 | | | | | |
| | | Hits@10 | 27.2 | | 53.6 | 16.0 | 7.3 | 39.2 | 46.7 | 26.2 | 21.1 | 24.0 | 11.0 | | | | | |
| | GNN-QE | Hits@3 | 38.1 | 22.7 | 58.6 | 24.5 | 22.3 | 53.0 | 62.1 | 39.0 | 28.4 | 33.4 | 21.7 | 26.4 | 24.5 | 13.6 | 20.2 | 28.5 |
| | | Hits@10 | 49.4 | 33.9 | 71.7 | 34.9 | 31.3 | 67.5 | 74.7 | 50.7 | 38.9 | 44.3 | 30.9 | 36.8 | 39.5 | 23.7 | 29.6 | 39.8 |
| 106% | Edge-type Heuristic | Hits@3 | 2.8 | 1.3 | 7.1 | 1.9 | 1.7 | 3.5 | 4.2 | 1.8 | 1.4 | 1.4 | 1.9 | 1.6 | 1.2 | 1.2 | 1.1 | 1.4 |
| | | Hits@10 | 6.4 | 3.0 | 14.5 | 4.3 | 4.4 | 8.1 | 9.7 | 4.8 | 3.7 | 3.1 | 4.7 | 2.8 | 2.2 | 4.4 | 2.6 | 2.9 |
| | NodePiece-QE | Hits@3 | 4.0 | | 11.9 | 3.6 | 4.0 | 3.6 | 4.0 | 1.8 | 2.1 | 1.5 | 3.3 | | | | | |
| | | Hits@10 | 7.7 | | 19.2 | 5.9 | 7.6 | 8.1 | 9.2 | 4.3 | 4.5 | 4.1 | 6.1 | | | | | |
| | NodePiece-QE w/ GNN | Hits@3 | 22.1 | | 39.6 | 14.8 | 5.1 | 30.1 | 35.6 | 19.6 | 19.6 | 24.5 | 9.9 | | | | | |
| | | Hits@10 | 31.7 | | 54.3 | 21.6 | 7.8 | 44.2 | 51.6 | 28.4 | 27.7 | 35.2 | 14.0 | | | | | |
| | GNN-QE | Hits@3 | 40.6 | 28.3 | 58.1 | 28.5 | 24.1 | 54.7 | 62.3 | 38.7 | 33.1 | 40.3 | 25.6 | 31.7 | 30.4 | 17.0 | 22.4 | 40.1 |
| | | Hits@10 | 50.4 | 37.7 | 71.9 | 37.0 | 32.4 | 67.9 | 73.7 | 48.2 | 41.8 | 48.0 | 32.7 | 39.6 | 43.2 | 25.3 | 31.1 | 49.4 |

Fig. 6 and Table 9 contain detailed results from the experiment in Section 5.3 about executing **training** queries over the original *training* and extended *test inference* graphs.

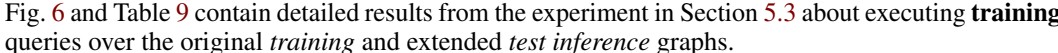

Figure 6: Hits@10 results on answering **training** queries executed over the original train (solid line) and test inference (dashed line) graphs. NodePiece-QE models are inference-only and were trained on *1p* queries, GNN-QE is end-to-end trainable on all complex queries.

Table 9: Hits@10 results (%) of **training** queries executed over the original *training* graph and extended *test inference* graph. All ratios $\mathcal{E}_{inf}/\mathcal{E}_{train}$. $\mathrm{avg}_p$ is the average on EPFO queries ($\wedge$, $\vee$). $\mathrm{avg}_n$ is the average on queries with negation. NodePiece-QE models are inference-only and were trained on *1p* queries, GNN-QE is end-to-end trainable on all complex queries.

| Ratio | Model | Graph | avg$_p$ | avg$_n$ | 1p | 2p | 3p | 2i | 3i | pi | ip | 2u | up | 2in | 3in | inp | pin | pni |
|---|---|---|---|---|---|---|---|---|---|---|---|---|---|---|---|---|---|---|
| 550% | Edge-type Heuristic | test | 25.4 | 10.5 | 19.2 | 21.3 | 23.4 | 36.1 | 29.4 | 18.9 | 23.6 | 27.3 | 29.2 | 7.0 | 4.6 | 23.1 | 10.4 | 7.4 |
| | NodePiece-QE | train | 79.1 | | 76.9 | 70.7 | 65.2 | 93.8 | 94.4 | 80.0 | 70.2 | 89.6 | 70.8 | | | | | |
| | | test | 48.2 | | 49.8 | 36.1 | 28.7 | 72.7 | 81.7 | 56.4 | 34.0 | 46.9 | 27.8 | | | | | |
| | NodePiece-QE w/ GNN | train | 80.0 | | 84.9 | 68.9 | 45.9 | 96.7 | 96.3 | 85.5 | 77.6 | 93.1 | 71.4 | | | | | |
| | | test | 55.7 | | 60.8 | 37.3 | 23.0 | 84.8 | 86.1 | 60.1 | 46.6 | 66.2 | 36.7 | | | | | |
| | GNN-QE | train | 99.9 | 99.7 | 100.0 | 99.9 | 99.8 | 100.0 | 100.0 | 100.0 | 100.0 | 100.0 | 99.8 | 99.7 | 100.0 | 99.6 | 99.6 | 99.8 |
| | | test | 95.4 | 92.1 | 99.5 | 94.8 | 83.9 | 99.9 | 100.0 | 99.1 | 93.0 | 99.3 | 89.5 | 95.8 | 98.1 | 87.4 | 87.3 | 92.1 |
| 300% | Edge-type Heuristic | test | 25.4 | 13.0 | 27.1 | 25.1 | 30.8 | 21.3 | 27.6 | 20.6 | 27.7 | 12.2 | 36.0 | 10.2 | 5.5 | 20.2 | 14.0 | 15.2 |
| | NodePiece-QE | train | 62.6 | | 68.3 | 55.3 | 54.6 | 68.3 | 76.0 | 62.2 | 58.5 | 58.2 | 62.0 | | | | | |
| | | test | 41.0 | | 51.8 | 30.1 | 28.4 | 56.3 | 66.5 | 42.1 | 30.7 | 31.7 | 31.1 | | | | | |
| | NodePiece-QE w/ GNN | train | 85.5 | | 96.9 | 81.3 | 49.9 | 97.5 | 98.2 | 89.5 | 86.0 | 91.4 | 78.7 | | | | | |
| | | test | 64.3 | | 82.2 | 51.9 | 33.1 | 86.4 | 90.5 | 67.8 | 55.9 | 62.8 | 47.9 | | | | | |
| | GNN-QE | train | 99.9 | 99.4 | 100.0 | 100.0 | 99.8 | 100.0 | 100.0 | 100.0 | 100.0 | 100.0 | 99.8 | 99.1 | 99.6 | 99.4 | 99.2 | 99.6 |
| | | test | 96.2 | 89.7 | 100.0 | 95.9 | 94.6 | 100.0 | 100.0 | 97.3 | 91.3 | 96.7 | 90.3 | 90.8 | 95.5 | 95.0 | 84.6 | 82.8 |
| 217% | Edge-type Heuristic | test | 21.9 | 11.1 | 24.3 | 22.2 | 24.6 | 20.8 | 30.0 | 21.4 | 18.9 | 9.3 | 25.6 | 7.4 | 9.9 | 16.0 | 10.3 | 12.1 |
| | NodePiece-QE | train | 59.7 | | 68.6 | 51.4 | 43.3 | 70.4 | 79.6 | 62.0 | 52.8 | 55.8 | 53.3 | | | | | |
| | | test | 45.0 | | 55.3 | 34.3 | 28.0 | 61.6 | 73.3 | 48.4 | 35.0 | 35.6 | 33.3 | | | | | |
| | NodePiece-QE w/ GNN | train | 83.9 | | 96.9 | 77.9 | 46.6 | 97.8 | 98.9 | 90.1 | 83.6 | 90.0 | 72.9 | | | | | |
| | | test | 71.0 | | 87.6 | 59.0 | 37.5 | 91.0 | 95.2 | 76.4 | 65.6 | 72.2 | 54.9 | | | | | |
| | GNN-QE | train | 99.9 | 98.7 | 100.0 | 99.9 | 99.7 | 100.0 | 100.0 | 100.0 | 99.9 | 100.0 | 99.8 | 98.6 | 99.2 | 98.7 | 98.3 | 98.5 |
| | | test | 98.3 | 93.9 | 100.0 | 97.6 | 96.9 | 100.0 | 100.0 | 99.0 | 96.8 | 99.6 | 96.2 | 95.5 | 95.6 | 96.1 | 91.8 | 90.5 |
| 175% | Edge-type Heuristic | test | 20.0 | 9.8 | 23.4 | 18.4 | 25.5 | 17.7 | 23.6 | 18.7 | 17.6 | 9.6 | 25.5 | 6.7 | 6.5 | 16.0 | 9.3 | 10.3 |
| | NodePiece-QE | train | 50.1 | | 65.8 | 42.0 | 40.7 | 54.2 | 62.1 | 47.1 | 43.1 | 49.0 | 47.0 | | | | | |
| | | test | 40.2 | | 57.6 | 30.8 | 29.1 | 47.9 | 57.4 | 38.6 | 31.8 | 35.5 | 33.0 | | | | | |
| | NodePiece-QE w/ GNN | train | 91.0 | | 99.9 | 94.1 | 51.7 | 99.9 | 99.9 | 96.3 | 96.8 | 98.4 | 82.2 | | | | | |
| | | test | 81.8 | | 95.8 | 76.2 | 43.4 | 97.3 | 98.7 | 87.4 | 84.3 | 88.5 | 64.6 | | | | | |
| | GNN-QE | train | 100.0 | 99.6 | 100.0 | 100.0 | 99.9 | 100.0 | 100.0 | 100.0 | 100.0 | 100.0 | 99.9 | 99.6 | 99.6 | 99.5 | 99.4 | 99.6 |
| | | test | 99.1 | 93.9 | 100.0 | 99.8 | 98.9 | 100.0 | 99.9 | 98.6 | 98.7 | 98.1 | 97.6 | 95.3 | 94.1 | 95.5 | 93.5 | 91.0 |
| 150% | Edge-type Heuristic | test | 19.7 | 9.8 | 27.0 | 19.1 | 24.8 | 16.6 | 23.6 | 17.6 | 15.9 | 8.4 | 24.7 | 6.2 | 7.7 | 15.1 | 9.2 | 10.6 |
| | NodePiece-QE | train | 41.3 | | 57.8 | 37.3 | 36.3 | 40.7 | 48.7 | 36.8 | 34.0 | 39.1 | 41.3 | | | | | |
| | | test | 34.8 | | 52.5 | 29.1 | 28.1 | 36.9 | 46.0 | 31.4 | 27.2 | 30.2 | 31.6 | | | | | |
| | NodePiece-QE w/ GNN | train | 88.5 | | 99.7 | 89.7 | 44.9 | 99.6 | 99.8 | 93.9 | 94.1 | 97.7 | 76.6 | | | | | |
| | | test | 79.6 | | 95.0 | 74.3 | 38.3 | 96.2 | 98.3 | 86.0 | 81.7 | 85.6 | 61.1 | | | | | |
| | GNN-QE | train | 99.9 | 98.8 | 100.0 | 99.9 | 99.8 | 100.0 | 100.0 | 99.9 | 99.9 | 100.0 | 99.9 | 98.6 | 98.8 | 98.9 | 98.7 | 99.1 |
| | | test | 98.8 | 94.1 | 100.0 | 99.6 | 98.5 | 99.9 | 99.9 | 99.2 | 98.4 | 96.5 | 97.4 | 94.6 | 95.5 | 94.9 | 92.9 | 92.3 |
| 133% | Edge-type Heuristic | test | 19.8 | 10.2 | 25.0 | 19.8 | 25.5 | 15.6 | 21.3 | 17.8 | 17.3 | 8.7 | 26.8 | 6.8 | 7.5 | 15.0 | 10.4 | 11.5 |
| | NodePiece-QE | train | 32.9 | | 52.0 | 32.0 | 34.1 | 25.0 | 27.7 | 26.0 | 29.9 | 31.9 | 37.4 | | | | | |
| | | test | 29.2 | | 49.0 | 27.5 | 29.4 | 22.9 | 26.0 | 23.0 | 25.8 | 27.0 | 32.1 | | | | | |
| | NodePiece-QE w/ GNN | train | 89.7 | | 99.9 | 92.0 | 47.6 | 99.9 | 99.9 | 94.1 | 96.0 | 99.2 | 78.8 | | | | | |
| | | test | 84.2 | | 97.3 | 82.0 | 43.0 | 97.8 | 98.9 | 89.3 | 88.5 | 92.4 | 69.0 | | | | | |
| | GNN-QE | train | 100.0 | 99.7 | 100.0 | 100.0 | 99.9 | 100.0 | 100.0 | 100.0 | 100.0 | 100.0 | 99.9 | 99.8 | 99.8 | 99.6 | 99.5 | 99.8 |
| | | test | 99.2 | 96.8 | 100.0 | 99.1 | 98.9 | 99.9 | 99.8 | 99.0 | 98.5 | 98.6 | 98.7 | 97.5 | 97.0 | 97.7 | 95.8 | 95.8 |
| 121% | Edge-type Heuristic | test | 17.9 | 8.6 | 22.9 | 16.6 | 23.9 | 14.6 | 20.7 | 15.8 | 14.0 | 7.9 | 24.3 | 5.4 | 5.9 | 13.8 | 8.6 | 9.2 |
| | NodePiece-QE | train | 38.4 | | 56.4 | 33.8 | 32.5 | 37.0 | 43.6 | 34.5 | 32.3 | 36.6 | 38.6 | | | | | |
| | | test | 35.0 | | 53.5 | 29.7 | 28.7 | 35.1 | 42.2 | 31.2 | 28.4 | 32.4 | 33.8 | | | | | |
| | NodePiece-QE w/ GNN | train | 87.8 | | 100.0 | 89.2 | 42.1 | 99.9 | 99.9 | 92.3 | 94.1 | 98.5 | 74.4 | | | | | |
| | | test | 84.8 | | 98.0 | 83.9 | 39.6 | 98.6 | 99.4 | 89.2 | 90.3 | 94.3 | 69.7 | | | | | |
| | GNN-QE | train | 100.0 | 99.4 | 100.0 | 100.0 | 99.9 | 100.0 | 100.0 | 100.0 | 100.0 | 100.0 | 99.9 | 99.4 | 99.5 | 99.1 | 99.3 | 99.7 |
| | | test | 99.3 | 97.3 | 100.0 | 99.2 | 99.2 | 100.0 | 100.0 | 99.3 | 99.3 | 98.6 | 98.6 | 97.9 | 98.3 | 97.8 | 96.5 | 96.2 |
| 113% | Edge-type Heuristic | test | 18.3 | 9.0 | 26.8 | 17.9 | 23.9 | 13.9 | 18.8 | 15.2 | 14.6 | 8.2 | 25.0 | 5.7 | 6.7 | 13.6 | 9.1 | 9.8 |
| | NodePiece-QE | train | 31.8 | | 52.5 | 29.4 | 30.3 | 27.0 | 30.5 | 25.5 | 26.7 | 30.8 | 33.9 | | | | | |
| | | test | 30.2 | | 51.0 | 27.6 | 28.4 | 25.9 | 29.6 | 24.1 | 25.0 | 28.6 | 31.8 | | | | | |
| | NodePiece-QE w/ GNN | train | 87.4 | | 100.0 | 88.4 | 40.4 | 99.9 | 99.9 | 91.9 | 94.0 | 99.6 | 72.4 | | | | | |
| | | test | 85.1 | | 98.8 | 83.9 | 38.6 | 99.0 | 99.5 | 89.9 | 90.7 | 97.2 | 67.9 | | | | | |
| | GNN-QE | train | 100.0 | 98.8 | 100.0 | 100.0 | 99.9 | 100.0 | 100.0 | 100.0 | 100.0 | 100.0 | 100.0 | 98.7 | 99.0 | 98.7 | 98.5 | 99.2 |
| | | test | 99.9 | 97.8 | 100.0 | 100.0 | 99.7 | 100.0 | 100.0 | 99.8 | 99.9 | 100.0 | 99.9 | 98.2 | 98.4 | 98.1 | 97.3 | 97.1 |
| 106% | Edge-type Heuristic | test | 17.1 | 8.3 | 24.1 | 16.1 | 23.6 | 13.2 | 17.4 | 14.4 | 14.4 | 7.8 | 23.9 | 5.4 | 5.5 | 13.5 | 8.2 | 9.0 |
| | NodePiece-QE | train | 24.1 | | 45.4 | 22.9 | 27.6 | 16.1 | 16.8 | 17.0 | 20.1 | 22.1 | 28.7 | | | | | |
| | | test | 23.6 | | 44.9 | 22.3 | 27.0 | 15.8 | 16.6 | 16.6 | 19.5 | 21.5 | 27.9 | | | | | |
| | NodePiece-QE w/ GNN | train | 86.0 | | 99.9 | 84.9 | 38.0 | 99.9 | 99.9 | 90.8 | 92.3 | 99.2 | 68.8 | | | | | |
| | | test | 84.9 | | 99.4 | 82.8 | 37.1 | 99.5 | 99.8 | 89.8 | 90.7 | 98.1 | 66.6 | | | | | |
| | GNN-QE | train | 100.0 | 99.0 | 100.0 | 100.0 | 99.9 | 100.0 | 100.0 | 100.0 | 99.9 | 100.0 | 99.9 | 98.8 | 99.0 | 98.9 | 98.9 | 99.4 |
| | | test | 99.9 | 98.4 | 100.0 | 100.0 | 99.8 | 100.0 | 100.0 | 100.0 | 99.8 | 100.0 | 99.7 | 98.4 | 98.5 | 98.3 | 98.1 | 98.8 |

# D Hyperparameters

Both NodePiece-QE and GNN-QE models are implemented with PyTorch [22] (MIT License). In particular, NodePiece-QE models employ PyG [12] (MIT License) and PyKEEN [2] (MIT License) for training link prediction models. GNN-QE is implemented based on the official NBFNet repository [7] (MIT License) and TorchDrug [41] library (Apache 2.0).

For all inductive experiments in Sections 5.2 and 5.3, Table 10 lists best hyperparameters for NodePiece-QE models without GNN encoder, Table 11 contains hyperparameters for GNN-enabled NodePiece-QE models. The GNN-free models use only relation-based tokenization where each entity $e$ is represented with two fixed-size sets: a set of $k$ unique *incoming* $r_i$ and a set of $k$ unique *outgoing* $r_o$ relation types. Looking up their $d$-dimensional vectors, we obtain:

$$e = \Big[ [\boldsymbol{r}_{i1}, \boldsymbol{r}_{i2}, \ldots, \boldsymbol{r}_{ik}][\boldsymbol{r}_{o1}, \boldsymbol{r}_{o2}, \ldots, \boldsymbol{r}_{ok}] \Big] \in \mathbb{R}^{2 \times k \times d}$$

If, for some entity, the number of unique relations of a certain kind is less than $k$, we pad the set with auxiliary [PAD] tokens. Entity representations are built as a function of the two sets $f(e) : \mathbb{R}^{2 \times k \times d} \to \mathbb{R}^d$:

$$\boldsymbol{h}_e = \text{MLP}\Big( \text{RANDOMPROJ}(\sum_{j=0}^{k} \boldsymbol{r}_{ij}) + \text{RANDOMPROJ}(\sum_{j=0}^{k} \boldsymbol{r}_{oj}) \Big)$$

Particularly, we first sum up tokens of the same direction, pass them through a random projection layer RANDOMPROJ (we found that making this projection learnable does not improve results), sum up representations of *incoming* and *outgoing* parts, and pass the resulting vector through a learnable MLP. This way, the number of learnable encoder parameters does not depend on the sequence length $k$, i.e., the number of chosen tokens per node.

The GNN-enabled models employ a slightly different *Concat + MLP* encoder where each node is tokenized with a sample of $k$ incident relations. Then, we concatenate $d$-dimensional embeddings of those *tokens* $t_i$ into a single long vector $\mathbb{R}^{kd}$, and then use a 2-layer MLP to project it to a model dimension $d$, i.e., $f(e) : \mathbb{R}^{kd} \to \mathbb{R}^d$:

$$\boldsymbol{h}_e = \text{MLP}\Big( [\boldsymbol{t}_0; \boldsymbol{t}_1; \ldots; \boldsymbol{t}_k] \Big)$$

For the large-scale experiment on WikiKG-QE in Section 5.5, we employ the *Concat + MLP* encoder. Instead of separating incoming and outgoing relation types, we first tokenize each node with 20 nearest *anchors* (pre-selected in advance using the default NodePiece strategy [13]) and add a sample of $k$ unique incident relations. NodePiece-QE w/ GNN employs a 3-layer CompGCN with the RotatE interaction function during message computation and *sum* aggregator. Due to high memory consumption, we train the 50d model for 2 epochs on 2 x RTX 8000 (48 GB) GPUs.

The overall tokens vocabulary consists of 20,000 anchor nodes, 1,024 relation types (including inverse relations) and one [PAD] token. All hyperparameters for this experiments are listed in Table 12.

Having trained the link predictors, we tune CQD-Beam hyperparameters on the validation set varying the t-norms, t-conorms, and scores normalization. Table 13 lists best options for each EPFO query type. For all experiments, we used a beam size $k = 32$ except for queries on WikiKG-QE where we used $k = 8$ due to the memory-expensive need of maintaining a beam over 2M entities.

Table 14 lists hyperparameters for GNN-QE models for all inductive splits. We found this architecture is quite stable under various configurations and eventually employed the same set of hyperparameters across all datasets.

BetaE (as a transductive baseline for the reference 175% dataset) was configured with $400d$ embedding dimension, batch size 512, 32 negative samples, learning rate 0.0005, margin $\gamma$ 60, and trained on 10 query patterns *{1p,2p,3p,2i,3i,2in,3in,inp,pin,pni}* for $200k$ steps.

---

[7]https://github.com/DeepGraphLearning/NBFNet

Table 10: NodePiece-QE hyperparameters for all inductive splits.

| Hyperparameter | Dataset $\mathcal{E}_{inf}/\mathcal{E}_{train}$ Ratios | | | | | | | | |
|---|---|---|---|---|---|---|---|---|---|
| | 106 | 113 | 133 | 134 | 150 | 175 | 217 | 300 | 550 |
| Vocab size | 472 | 466 | 460 | 458 | 450 | 438 | 442 | 402 | 346 |
| Tokens per node | 20 | 20 | 20 | 20 | 20 | 20 | 20 | 20 | 10 |
| Vocab dim | 400 | 400 | 400 | 400 | 400 | 400 | 400 | 400 | 1000 |
| Scoring function | ComplEx [30] | | | | | | | | |
| Encoder | RandomProj + MLP | | | | | | | | |
| Encoder dim | 400 | 400 | 400 | 400 | 400 | 400 | 400 | 400 | 1000 |
| Encoder layers | 2 | | | | | | | | |
| Batch size | 256 | | | | | | | | |
| Epochs | 400 | 400 | 400 | 1000 | 1000 | 2000 | 2000 | 2000 | 3000 |
| Learning rate | 1e-4 | | | | | | | | |
| Optimizer | Adam | | | | | | | | |
| Loss function | BCE | | | | | | | | |
| Adv. temperature | 1.0 | 1.0 | 1.0 | 1.0 | 0.5 | 0.5 | 0.5 | 0.2 | 1.0 |
| # negatives | 128 | 128 | 128 | 128 | 128 | 128 | 128 | 128 | 128 |
| # parameters | 699k | 694k | 689k | 688k | 681k | 671k | 675k | 643k | 2.7M |
| Training time (hrs) | 15 | 12 | 10 | 9 | 6 | 9 | 9 | 5 | 1 |

Table 11: NodePiece-QE with GNN hyperparameters for all inductive splits.

| Hyperparameter | Dataset $\mathcal{E}_{inf}/\mathcal{E}_{train}$ Ratios | | | | | | | | |
|---|---|---|---|---|---|---|---|---|---|
| | 106 | 113 | 133 | 134 | 150 | 175 | 217 | 300 | 550 |
| Vocab size | 472 | 466 | 460 | 458 | 450 | 438 | 442 | 402 | 346 |
| Tokens per node | 20 | | | | | | | | 10 |
| Vocab dim | 200 | | | | | | | | 400 |
| Scoring function | ComplEx [30] | | | | | | | | |
| Encoder | Concat + MLP | | | | | | | | |
| Encoder dim | 200 | | | | | | | | 400 |
| Encoder layers | 2 | | | | | | | | |
| GNN encoder | CompGCN [31] + RotatE [27] message function | | | | | | | | |
| GNN layers | 5 | 5 | 5 | 5 | 5 | 3 | 3 | 3 | 3 |
| GNN dim | 200 | | | | | | | | 400 |
| Batch size | 256 | | | | | | | | |
| Epochs | 600 | 1000 | 1000 | 1000 | 1000 | 1000 | 3000 | 4000 | 4000 |
| Learning rate | 5e-4 | | | | | | | | |
| Optimizer | Adam | | | | | | | | |
| Loss function | BCE | | | | | | | | |
| Adv. temperature | 1.0 | 1.0 | 1.0 | 1.0 | 1.0 | 1.0 | 1.0 | 0.5 | 1.0 |
| # negatives | 128 | 128 | 128 | 128 | 128 | 128 | 128 | 128 | 128 |
| # parameters | 2.8M | 2.8M | 2.8M | 2.8M | 2.8M | 2.4M | 2.4M | 2.3M | 5.7M |
| Training time (hrs) | 30 | 30 | 19 | 20 | 14 | 15 | 8 | 5 | 1 |

Table 12: NodePiece-QE hyperparameters for WikiKG-QE (133%).

| Hyperparameter | NodePiece-QE | NodePiece-QE w/ GNN |
|---|---|---|
| Vocab size | 20,000 anchors + 1024 relation types | |
| Anchor tokens per node | 20 | 15 |
| Relation tokens per node | 20 | 10 |
| Vocab dim | 100 | 50 |
| Scoring function | ComplEx [30] | |
| Encoder | Concat + MLP | |
| Encoder dim | 200 | 100 |
| Encoder layers | 2 | 2 |
| GNN | - | CompGCN |
| GNN dim | - | 50 |
| GNN layers | - | 3 |
| Batch size | 512 | 1024 |
| Epochs | 40 ($\approx$ 1M steps) | 2 |
| Learning rate | 1e-4 | |
| Optimizer | Adam | |
| Loss function | BCE | |
| Adversarial temp. | 1.0 | |
| # negatives | 64 | 512 |
| # parameters | 2,922,900 | 1,211,950 |
| Training time (hrs) | 40 | 16 |

Table 13: CQD-Beam t-norm hyperparameters for all splits and both link predictors, NodePiece-QE and NodePiece-QE w/ GNN, when answering EPFO queries. The default beam size $k = 32$, *prod +* $\sigma$ is product t-norm with sigmoid score normalization. Details on t-norms are in Appendix A.

| Ratio | Link predictor | 1p | 2p | 3p | 2i | 3i | pi | ip | 2u | up |
|---|---|---|---|---|---|---|---|---|---|---|
| 106% | NodePiece-QE
NodePiece-QE w/ GNN | prod + $\sigma$ | prod + $\sigma$ | prod + $\sigma$ | prod + $\sigma$ | prod + $\sigma$ | prod + $\sigma$ | prod + $\sigma$ | prod + $\sigma$ | prod + $\sigma$ |
| 113% | NodePiece-QE
NodePiece-QE w/ GNN | prod + $\sigma$ | prod + $\sigma$ | prod + $\sigma$ | prod + $\sigma$ | prod + $\sigma$ | prod + $\sigma$ | prod + $\sigma$ | prod + $\sigma$ | prod + $\sigma$ |
| 122% | NodePiece-QE
NodePiece-QE w/ GNN | prod + $\sigma$ | prod + $\sigma$ | prod + $\sigma$ | prod + $\sigma$ | prod + $\sigma$ | prod + $\sigma$ | prod + $\sigma$ | prod + $\sigma$ | prod + $\sigma$ |
| 134% | NodePiece-QE
NodePiece-QE w/ GNN | prod + $\sigma$ | prod + $\sigma$ | prod + $\sigma$ | prod + $\sigma$ | prod + $\sigma$ | prod + $\sigma$ | prod + $\sigma$ | prod + $\sigma$ | prod + $\sigma$ |
| 150% | NodePiece-QE
NodePiece-QE w/ GNN | prod + $\sigma$ | prod + $\sigma$ | prod + $\sigma$ | prod + $\sigma$ | prod + $\sigma$ | prod + $\sigma$ | prod + $\sigma$ | prod + $\sigma$ | prod + $\sigma$ |
| 175% | NodePiece-QE
NodePiece-QE w/ GNN | prod + $\sigma$ | prod + $\sigma$ | prod + $\sigma$ | prod + $\sigma$ | prod + $\sigma$ | prod + $\sigma$ | prod + $\sigma$ | prod + $\sigma$ | prod + $\sigma$ |
| 217% | NodePiece-QE
NodePiece-QE w/ GNN | prod + $\sigma$ | prod + $\sigma$ | prod + $\sigma$ | prod + $\sigma$ | prod + $\sigma$ | prod + $\sigma$ | prod + $\sigma$ | prod + $\sigma$ | prod + $\sigma$ |
| 300% | NodePiece-QE
NodePiece-QE w/ GNN | prod + $\sigma$ | prod + $\sigma$ | prod + $\sigma$ | prod + $\sigma$ | prod + $\sigma$ | prod + $\sigma$ | prod + $\sigma$ | prod + $\sigma$ | prod + $\sigma$ |
| 550% | NodePiece-QE
NodePiece-QE w/ GNN | prod + $\sigma$ | prod + $\sigma$ | prod + $\sigma$ | prod + $\sigma$ | prod + $\sigma$ | prod + $\sigma$ | prod + $\sigma$ | prod + $\sigma$ | prod + $\sigma$ |
| 133% | NodePiece-QE | (WikiKG-QE) prod + $\sigma$ for all query types | | | | | | | | |

Table 14: Hyperparameters of GNN-QE on different datasets. All the hyperparameters are selected by the performance on the validation set.

| Hyperparameter | | All splits |
|---|---|---|
| **GNN** | #layers | 4 |
| | hidden dim. | 32 |
| | composition | DistMult [33] |
| | aggregation | PNA [10] |
| **MLP** | #layer | 2 |
| | hidden dim. | 64 |
| **Traversal Dropout** | probability | 0.5 |
| **Logical Operator** | t-norm | product |
| **Learning** | batch size | 64 |
| | sample weight | uniform across queries |
| | loss | BCE |
| | # negatives | 32 |
| | optimizer | Adam |
| | learning rate | 5e-3 |
| | iterations (#batch) | 10,000 |
| | adv. temperature | 0.1 |

# E    Edge-type Heuristic

We consider *Edge-type Heuristic* as a trivial baseline for inductive complex query. Given a query $\mathcal{Q} = (\mathcal{C}, \mathcal{R}_Q, \mathcal{G})$, Edge-type Heuristic finds all entities $e \in \mathcal{E}$ that satisfy the relations in the last hop of $\mathcal{R}_Q$ on the inference graph $\mathcal{G}_{inf}$. In other words, this baseline filters out entities that are not consistent with the query according to the edge types, which is a necessary condition for the answers when the inference graph is reasonably dense. Since Edge-type Heuristic only distinguishes the entities into two classes, we randomly shuffle the entities in each class to create a ranking.

Edge-type Heuristic can be easily implemented as GNN-QE with a special relation *projection*. Given an inference graph $\mathcal{G}_{inf}$, we first preprocess a relation-to-entity mapping $M$, where $M_{r,t}$ means there exists a head entity $h$ and an edge $(h, r, t)$ for tail entity $t$. Then the relation *projection* of relation $r$ can be implemented by looking up the row $M_r$. Note that the Edge-type Heuristic only filters entities according to the edge type, and hence the head entity (or the distribution of head entity) is ignored in the relation projection.