# OpenReview forum: "Inductive Logical Query Answering in Knowledge Graphs"
_NeurIPS.cc/2022/Conference — NeurIPS 2022 Accept_

### Official Review · Reviewer_YfZT · 2022-07-10

**Rating:** 5
**Confidence:** 5
**Soundness:** 3 good
**Presentation:** 4 excellent
**Contribution:** 3 good

**Summary:**

In this paper, the authors studied the problem of logical query answering over knowledge graphs under an inductive setting. Existing methods usually focus on the transductive setting where all nodes and edges are seen during the training stage. This work goes beyond and investigates the scenario where there are unseen nodes in the validation and testing stages. To tackle this problem, they proposed two methods and demonstrated their effectiveness via extensive experiments.

**Questions:**

- Since the node representation is learned using the edge structural patterns. I am wondering if the model will output the same representations for different nodes (which I think is not ideal).
- Is there any way to marry the benefits of both models?


**Limitations:**

Although the NBFNet-QE can achieve good performance, it is not scalable, and no results are reported for the WikiKG dataset. However, the more efficient NodePiece-QE performs rather worse. This limitation makes me doubt the usability of the proposed two approaches.

**Strengths And Weaknesses:**

### Strength
- The studied problem is novel and interesting. The inductive setting is novel and more generalized.
- They set up new benchmarks for the inductive logical query task by constructing two new datasets, one using FB15K-237 and a large one using OBG WikiKG.
- The proposed NBFNet-QE performs well on the new FB15K-237 dataset (The numbers are relatively higher compared to the transductive setting) and outperforms NodePiece-QE by a large margin. This finding is interesting and may provide directions for future improvement.
- The presentation is clear and easy to follow.

### Weakness
- The contribution to the model design is rather incremental. Both models (NodePiece and NBFNET) are based on existing works on link prediction (a simpler yet closely related task). The query answering part is also based on existing transductive logical query answering models  (e.g., CQD) on KG.
- The proposed framework is only applicable to unseen nodes as the node representation learning process heavily depends on the edges (relations). It cannot deal with unseen relations.
- The discussion on the efficiency-effectiveness trade-off between NodePiece-QEand NBFNet-QE is a bit trivial. Solutions/insights that can address this trade-off are expected.

---

> ### Author Response · Authors · 2022-08-02
> **Response to Reviewer YfZT, Part 1**
>
> We thank the Reviewer for acknowledging the novelty of the studied problem, experimental setup, and presentation of the manuscript.
> Below, we would like to comment on the weaknesses and limitations.
>
> > **W1. Both models are based on existing works on link prediction … The contribution to the model design is rather incremental.**
>
> Although the proposed baseline models are adaptations of published works to the inductive setup, we would like to emphasize that we did not claim any contribution as to the model design in this work. Instead, we aim at designing a principled approach towards studying the inductiveness (and defining the inductive logical query answering task) and possible ways to achieve it with existing graph representation learning methods that we categorize into inductive node representations and inductive relational structure representations. Choosing (or baking into a model) a suitable inductive bias is still an open challenge in representation learning and it is often accompanied with certain pros and cons on which we focused in this work.
>
> In the paragraph describing contributions (lines 61-69) we introduce the new inductive complex query answering setup and important challenges of those models such as OOD generalization to larger inference graphs, scaling to large graphs, and the ability to recover new answers for train queries. Solving some of those challenges (eg, OOD generalization) is an ongoing effort of the Graph ML community, and, of course, we could not tackle all of them in a single piece of work proposing the “best model” that would immediately solve the task. Nevertheless, we believe that the proposed baselines demonstrate a non-trivial performance and would be strong competitors in their respective branches (such inference-only models and end-to-end trainable).
>
> > **W2. The proposed framework is only applicable to unseen nodes as the node representation learning process heavily depends on the edges (relations). It cannot deal with unseen relations.**
>
> (This issue is also brought up by Reviewer tzzT.) Albeit it is correct that the set of relations has to be fixed, we would not attribute this fact as a weakness of this particular work, but rather as a weakness (or rather a deliberate design choice) of the whole inductive link prediction literature with its influential examples like GraIL [1], RED-GNN [2], and NBFNet [3].
>
> We follow that formulation of the inductive task in Section 3 and assume that the set of relation types remains the same. The problem of inductive reasoning over *unseen relations* is highly complex and deserves a dedicated research paper - to the best of our knowledge, this setup has not been studied on theoretical or practical (like simple link prediction) levels in recent ML venues. This is definitely a solid avenue for a future work.
>
> > **W3. The discussion on the efficiency-effectiveness trade-off between NodePiece-QE and NBFNet-QE is a bit trivial. Solutions/insights that can address this trade-off are expected + This limitation makes me doubt the usability of the proposed two approaches + Is there any way to marry the benefits of both models?**
>
> We could not delve into this discussion due to space limitations, but we see at least five avenues towards bridging the trade-off.
>
> (A1) Improved pre-training of inference-only models. Please find in the revised version of the manuscript that we have updated the performance of NodePiece-QE w/ GNN on all datasets thanks to a better training strategy with almost 2X average EPFO improvement. With that, the gap between the inference-only NodePiece-QE w/ GNN and trainable NBFNet-QE is significantly reduced, and in some queries and dataset ratios, the inference-only model even outperforms the trainable model, e.g., in *ip* and *2u* queries (Figure 5 in Appendix C). We also observe an improved performance of NodePiece-QE w/ GNN on training queries. We believe this is a strong signal that better fitting of a simple link prediction model might result in significant performance gains (we briefly touch upon this fact in lines 292-294 in Section 5.3 of the manuscript but will include a more detailed discussion in the camera-ready version)

---

> > ### Author Response · Authors · 2022-08-02
> > **Response to Reviewer YfZT, Part 2**
> >
> > (A2) Better scaling of relational GNNs especially in terms of GPU memory consumption. As discussed in the answer to Reviewer YzXG pertaining to the WikiKG dataset, GraphSAGE-style neighborhood sampling won’t help much in the link prediction setup as graph diameter is relatively small, and with a high number of negative samples, the input edge index quickly becomes comparable to the full graph with 3- and higher hops neighborhood sampling. Besides, at inference time, we still have to rank all 2M nodes and it is still more efficient to run a GNN in the full-batch mode. That said, during the rebuttal period, we managed to pre-train a 50d NodePiece-QE w/ GNN in the full-batch mode for 1 epoch on a 48 GB GPU (which takes about 20 hours) and reported initial results - the 1p performance is surprisingly high while complex queries performance is quite low. We commit to running pre-training in the distributed mode for a longer time and report updated numbers in the camera-ready version.
> >
> > (A3) Distillation of trained NBFNet-QE to more efficient shallow node embeddings. This is a rather non-trivial task as it implies distilling a model of one architecture to a totally different (or simpler) one. But that would theoretically allow to run a model with performance like NBFNet-QE in the inference-only mode.
> >
> > (A4) More expressive decoding strategies for inference-only methods. We observe from the experiments that for CQD-Beam reaching high 1p simple link prediction pre-training accuracy does not immediately correspond to improving the performance on complex queries. For example, increasing 1p accuracy by 10 Hits@10 points does not correspond to increasing the accuracy on 2p, 3p, and other queries by 10 points. We hypothesize it might happen due to a simple pointwise scoring function (when processing a projection atom query $(h, r, ?)$) over entity vectors. We hypothesize that incorporating the graph structure into the decoding process might further improve the prediction quality.
> >
> > (A5) Alleviating the OOD generalization issue of GNNs when running inference on larger graphs than a model saw during training. This is a known issue of GNNs such that any progress in this problem would lead to increases in both inference-only and trainable models. For example, a recent work by Zhou et al [4] proposes some theoretical and practical insights how to imbue GNNs with better generalization capabilities but at the cost of increased computational complexity. We believe this is a promising area for new inductive GNN architectures.
> >
> > We will add those discussions in the final version.
> >
> > > **Q1. Since the node representation is learned using the edge structural patterns. I am wondering if the model will output the same representations for different nodes (which I think is not ideal).**
> >
> > For NBFNet-QE, the model *might* return the same representations for two nodes if they form an automorphism (including relation types). It is theoretically possible but very rarely happens in any real-world non-regular graphs such as our sampled datasets. We would further hypothesize that if two nodes do form an automorphism, it then does make sense to predict links to both of them.
> >
> > For NodePiece-QE, two nodes tokenized with the same sequence of incident relations and/or anchors will have the same initial encoded representation (akin to hash collision). The tokenization procedure will return the same tokens for two nodes only of they share exactly the same set of unique incident relations (and top-k nearest anchors if using a vocabulary of anchors). However, in our datasets we observe that more than 90% of nodes have unique tokenizations which leads to unique initial encoded representations.
> >
> > We can further alleviate the issue of exactly the same node representations by adding any message passing GNN on top of NodePiece hashes, i.e., it would work akin to a 1-WL kernel that assumes all input features are the same and after several message passing steps it often returns unique node representations (subject to isomorphism, but, again, in any large non-regular graphs this is practically rare). That is, a GNN enriches initial NodePiece representations by aggregating neighboring features and makes features more distinguishable. In practice, this hypothesis holds as we see experimentally a better performance of NodePiece-QE w/ GNN compared to a plain NodePiece-QE.
> >
> > References:
> > [1] Teru et al. Inductive Relation Prediction by Subgraph Reasoning. ICML 2020
> > [2] Zhang and Yao. Knowledge Graph Reasoning with Relational Digraph. WWW 2022
> > [3] Zhu et al. Neural Bellman-Ford Networks: A General Graph Neural Network Framework for Link Prediction. NeurIPS 2021
> > [4] Zhou et al. OOD Link Prediction Generalization Capabilities of Message-Passing GNNs in Larger Test Graphs. arxiv:2205.15117

---

### Official Review · Reviewer_PdEh · 2022-07-11

**Rating:** 6
**Confidence:** 4
**Soundness:** 3 good
**Presentation:** 3 good
**Contribution:** 3 good

**Summary:**

This work studies the inductive query answering task where inference is performed on a graph containing new entities with queries over both seen and unseen entities. Which is a sound contribution to the field of query answering and knowledge graph completion.

The authors argue that several existing studies on complex query answering with logical operators have been conducted in transductive mode where the set of entities does not change at inference time.

The aim of this study is to extend the complex logical query answering to inductive representation learning algorithms. Focusing on the inductive setup where an inference graph is a superset of a training graph such that:
- Inference queries require reasoning over both seen and new entities;
- Original training queries might have more correct answers at inference time with  the addition of new entities.

In this work;
- They show that inductive models are able to perform logical reasoning at inference time over unseen nodes generalizing to graphs up to 500% larger than training ones.
- They also explore the efficiency-effectiveness trade-off, and the findings indicate that the inductive relational structure method generally achieves higher performance.
- Also the inductive node representation method is able to answer complex queries in the inference regime only without any training on queries and scale to graphs of millions of nodes.


**Questions:**

N/A

**Strengths And Weaknesses:**

- The paper extends the idea of inductive representation learning algorithms to logical query answering where inference queries require reasoning over both seen and new entities which is a sound contribution to the field of query answering and knowledge graph completion.
- The authors also cite/explain the related works in both knowledge graph completion and complex query answering. The paper also makes a good differentiation between the proposed approach and the existing work.

- The paper is well written, with elaborative figures and experimental results support the proposed method.

---

> ### Author Response · Authors · 2022-08-02
> **Response to Reviewer PdEh**
>
> We thank the Reviewer for acknowledging the soundness of our contribution to the field of query answering and KG completion, its positioning among related work, and overall clarity of the manuscript.
> We would like to ask whether you have any further question so we might convince you to increase the final score?

---

> > ### Comment · Reviewer_PdEh · 2022-08-08
> > **Thank you for responding.**
> >
> > Thanks for taking the time to respond. I don't have any more queries. I do appreciate your work and the contribution. In light of your responses, I am still comfortable sticking to my current score.

---

### Official Review · Reviewer_YzXG · 2022-07-12

**Rating:** 8
**Confidence:** 4
**Soundness:** 4 excellent
**Presentation:** 4 excellent
**Contribution:** 3 good

**Summary:**

This work proposes the task of Inductive Logical Query Answering which is the natural next step after past work on:
* KG Completion: Predicting new triples in a knowledge graph (KG) with a fixed set of entities
* Inductive KG Completion: Predicting triples for new entities (and some of their triples) added to an existing KG
* Logical Query Answering: Executing First-Order Logic (FOL) queries on a KG where we assume that the KG is incomplete and the set of answers contains more entities than those that can be predicted from just FOL query execution

Inductive Logical Query Answering is then executing FOL queries in a KG where new entities are added at inference time. The inductive settings are motivated by real-world scenarios where KGs are constantly being updated with new information and new entities.

The authors then propose two approaches (each with its own strengths and weaknesses) to solve the task.

**Questions:**

* Line 208: Can you flesh out how the mechanism of NBFNet is a *labeling trick*?
* This is somewhat implied from the text. Are the validation and test queries guaranteed to require missing edges to find the full answer set? Does the query sampling ensure this?
* How do you explain the initial increase in performance as the graph grows (e.g. in Fig 3)?
* Line 347: Performance in Table 2 on the large 2M node graph shows the accuracy of <10%. Why is this considered non-trivial? Is trivial just considered random guessing performance? There should be better baseline estimates that can scale to this graph size with graph neighborhood sampling.
* Following up on the discussion of better evaluation metrics, how many times does the system predict a "hard" answer higher than an "easy" answer? Is this a strict constraint that should be applied when evaluating systems on these tasks?

**Limitations:**

The weakness section highlights a missing discussion on the evaluation of systems in the inductive setting. This is a limitation of the query answering task in general and not just of this work. However, given that this work adds an additional component of inductive KG growth, this discussion seems relevant and necessary for completeness.

**Strengths And Weaknesses:**

Strengths

[+] Inductive query answering is well-motivated as an extension of past work. This work is also well-placed with respect to related work.

[+] The paper is well-presented and descriptions are mostly sufficient to understand the work.

[+] The baseline approaches described are suitable for the task and have complementary strengths and weaknesses.

Weaknesses

[-] There should be more discussion on evaluating FOL execution systems in the inductive setting and what the right metric for the task is.

Reporting accuracy of the systems on train queries at inference is an important first step toward establishing "faithfulness". Can we trust a system that does not achieve 100% accuracy on "easy" answers?

Similarly, just reporting Hits@10 (after filtering out "Easy" answers) hides under the rug what it means to predict a "hard" answer at a higher rank than an "easy" answer (before filtering). Personally, I am of the opinion that the evaluation metric should penalize any entity being predicted as an answer before all the "easy" answers are enumerated. This ensures trust in the system to first execute the FOL query when possible and only then resort to "guessing" more answers by filling in missing edges.

---

> ### Author Response · Authors · 2022-08-02
> **Response to Reviewer YzXG, Part 1**
>
> We thank the Reviewer for acknowledging the novelty and presentation of the manuscript.
> Below, we would like to comment on the weaknesses and answer the questions.
>
> > **There should be more discussion on evaluating FOL execution systems in the inductive setting and what the right metric for the task is… Can we trust a system that does not achieve 100% accuracy on "easy" answers?**
>
> We agree that existing evaluation protocols might be far from perfect - in this work, we mostly follow the transductive query answering literature, i.e., seminal works Query2Box [1] and BetaE [2] who employed link prediction metrics MRR and Hits@k in evaluation, and the community decided to stick to that. The problem of logical query answering is definitely more challenging than simple link prediction and we believe it is a reasonable assumption that MRR / Hits@k might not capture all characteristics of a more complex task.
>
> That said, we observe that more refined metrics started to appear, e.g., “faithfulness” introduced by EmQL [3] as an ability to recover easy answers. In the inductive setup where the inference graph extends the training one, we extend the notion of “faithfulness” and measure whether query answering models are able to identify new “easy” answers to known training queries by traversing newly added entities and edges (no link prediction required here and no “hard” answers) - this is the experiment reported in Section 5.3.
>
> Generally, we are inclined to trust predictions of a system that does not perfectly recover all easy answers -  some models / inductive biases / hyperparameters might work better in generalization than entailment. Still, it is always possible to equip a well-generalizable model with a purely symbolic model (edge traversal) that would perfectly solve easy answers but would be rather useless for hard answers.
>
> > **the evaluation metric should penalize any entity being predicted as an answer before all the "easy" answers are enumerated … how many times does the system predict a "hard" answer higher than an "easy" answer?**
>
> Indeed, measuring whether all “easy” answers can be found before guessing “hard” ones makes a lot of sense - we believe ROC AUC over **unfiltered** predictions including both easy and hard answers fits here the best. We performed an inference experiment on the reference 175% dataset (the trained models reported in Table 1) computing ROC AUC of unfiltered predictions.
>
> The idea of computing ROC AUC here is as follows:
> Suppose we have a list of unfiltered raw scores from which we extract scores of all easy and hard answers. Suppose we have a query with 4 easy and 1 hard answer:
> ```
> [5, 6, 7, **8**, 32]  # 8 is a rank of a hard answer while 5, 6, 7, 32 are ranks of easy answers
> ```
> We then create binary labels for the scores assigning 1 to the hard answers, e.g.
> ```
> [0, 0, 0, 1, 0]
> ```
> Given those two arrays, we then compute the ROC AUC score that would measure how many hard answers are ranked after easy answers, e.g., in our example ROC AUC is 0.75. Note that the score  does not depend on actual values of ranks, that is, the metric will be high when easy answers are eg ranked 1000-1004 as long as hard answers are ranked 1005 and lower. So ideally we would still need to pair ROC AUC with MRR to see where easy and hard answers are ranked absolutely.
>
> We compute ROC AUC for each query and average them over each query type thus making it **macro-averaged ROC AUC** (we note that queries with few answers and one-two misplaced hard predictions give a low score, so ideally we’d need some normalization by the answer set size). Our experimental results on all query types on the reference 175% dataset:
>
> |                     | avg_p  | avg_n   | 1p    | 2p    | 3p    | 2i    | 3i    | pi    | ip    | 2u    | up    | 2in   | 3in   | inp   | pin   | pni   |
> | ------------------- | ----- | ----- | ----- | ----- | ----- | ----- | ----- | ----- | ----- | ----- | ----- | ----- | ----- | ----- | ----- | ----- |
> | NodePiece-QE        | 0.653 |       | 0.609 | 0.656 | 0.666 | 0.628 | 0.627 | 0.645 | 0.681 | 0.677 | 0.686 |       |       |       |       |       |
> | NodePiece-QE w/ GNN | 0.675 |       | 0.688 | 0.689 | 0.667 | 0.655 | 0.624 | 0.646 | 0.69  | 0.713 | 0.701 |       |       |       |       |       |
> | NBFNet-QE           | 0.978 | 0.901 | 0.997 | 0.981 | 0.962 | 0.987 | 0.978 | 0.975 | 0.975 | 0.982 | 0.965 | 0.919 | 0.884 | 0.888 | 0.913 | 0.904 |
>
> NBFNet-QE performs almost perfectly w.r.t. ROC AUC as it was trained on complex queries. NodePiece-QE models are pretty good for inference-only models that were only trained only on 1p simple link prediction and have never seen any complex query at training time.  We added this experiment to the Appendix E of the updated manuscript and will add more ROC AUC results for other dataset splits in the camera-ready version.

---

> > ### Author Response · Authors · 2022-08-02
> > **Response to Reviewer YzXG, Part 2**
> >
> > > **Can you flesh out how the mechanism of NBFNet is a labeling trick?**
> >
> > Sure, the idea of the labeling trick [4] assumes that certain nodes of interest have some unique “label” (often expressed as an additional feature vector) to break neighborhood symmetries that might end up with nodes having the same final updated vector.
> > NBFNet is designed for the link prediction task, particularly, for answering simple queries $(h, r, ?)$. For *each* training or inference query, NBFNet initializes a graph with **unique** starting node features - for instance, for a $(h, r, ?)$ query, one single node $h$ is initialized with a “query” vector corresponding to relation $r$ while **all other** nodes in a graph are initialized with all-zero vectors (edges in the graph retain their relation type features). Therefore, we uniquely label a node $h$ with a unique query vector and perform message passing through NBFNet layers to get final node states and a distribution over all nodes as possible answers - we denote it as a *conditional distribution* $p(t|h,r)$ since the predicted tail node $t$ is conditioned on the unique initialization of head node $h$ and query relation $r$. The labeling trick is powerful but computationally expensive, e.g., at inference, we need a separate GNN forward pass over the whole graph for each query.
> >
> > We’d add that the query embedding matrix is not the same as edge features used in the NBFNet layer - the edge features are obtained either via a layer-specific linear projection $f: \mathcal{R}^d \rightarrow \mathcal{R}^{|R| \times d}$ that maps from a query vector to a matrix of all relation types, or via a layer-specific learnable embedding matrix of all edge types. For example, in a 3-layer NBFNet we would have 1 $\mathcal{R}^{|R| \times d}$ query matrix and 3 more relation embedding matrices of the same shape, one per layer.
> >
> > > **Are the validation and test queries guaranteed to require missing edges to find the full answer set? Does the query sampling ensure this?**
> >
> > Yes, it is guaranteed by the query sampling procedure that validation and test queries have at least one missing edge to predict at inference time. Without predicting this missing edge, a model won’t be able to recover “hard” answers and, therefore, the full answer set.
> >
> > > **How do you explain the initial increase in performance as the graph grows (e.g. in Fig 3)?**
> >
> > Please note that we have updated the manuscript with new results of NodePiece-QE w/ GNN.
> > Generally, we observe that GNN-based models like NodePiece-QE w/ GNN and NBFNet-QE are more susceptible to this performance bump. This performance bump phenomenon is indeed interesting and we definitely plan to investigate it further to make sure it is not just a dataset noise. So far, we hypothesize the final performance on larger inductive graphs is affected by two components: (1) positively by reducing the average node degree in the inference graphs while the number of unique relations stays more or less the same (440-470 relations, more stats in Table 3 in Appendix B), such that edge types are more “informative“; and (2) negatively by a known fact that GNNs have troubles generalizing to graphs larger than training ones - a recent work by Zhou et al [5] shows interesting theoretical reasons for that. In simpler words, we hypothesize that the performance grows as long as the positive component outweighs the negative effect of OOD generalization, and eventually all GNN-based models deteriorate as the negative effect prevails. We will include this discussion in the camera-ready version.

---

> > > ### Author Response · Authors · 2022-08-02
> > > **Response to Reviewer YzXG, Part 3**
> > >
> > > > **Performance in Table 2 on the large 2M node graph shows the accuracy of <10%. Why is this considered non-trivial? Is trivial just considered random guessing performance? There should be better baseline estimates that can scale to this graph size with graph neighborhood sampling.**
> > >
> > > Yes, the performance is non-trivial w.r.t random guessing - the probability of an answer to be in top-100 predictions over 2M nodes is $100 / 2*10^6  = 0.00005$. We would like to emphasize the hardness of this large-scale task, i.e., at inference time a model sees 500k new nodes and 4M new edges, and we perform ranking against all 2M nodes for each query.
> > > Please note that we have updated Table 2 in the manuscript with (i) new numbers for NodePiece-QE improving the average EPFO H@100 from 6.6 to 9.2 thanks to a better pre-training strategy as well as (ii) added a preliminary version of a full-batch NodePiece-QE w/ GNN.
> > >
> > > For any relational GNN, GraphSAGE-style neighborhood sampling won’t help much here (most of sampling approaches were created for node classification) in the link prediction setup as graph diameter is relatively small, and with a high number of negative samples, the input edge index quickly becomes comparable to the full graph with 3- and higher hops neighborhood sampling. Besides, at inference time, we still have to rank all 2M nodes and it is still more efficient to run a GNN in the full-batch mode. That said, during the rebuttal period, we managed to pre-train a 50d NodePiece-QE w/ GNN in the full-batch mode for 1 epoch on a 48 GB GPU (which takes about 20 hours) and reported initial results - the 1p performance is surprisingly high while complex queries performance is quite low. We commit to running pre-training in the distributed mode for a longer time and report updated numbers in the camera-ready version.
> > >
> > >
> > > [1] Ren et al. Query2box: Reasoning over Knowledge Graphs in Vector Space using Box Embeddings. ICLR 2020
> > > [2] Ren and Leskovec. Beta Embeddings for Multi-Hop Logical Reasoning in Knowledge Graphs. NeurIPS 2020
> > > [3] Sun et al. Faithful Embeddings for Knowledge Base Queries. NeurIPS 2020
> > > [4] Zhang et al. Labeling Trick: A Theory of Using Graph Neural Networks for Multi-Node Representation Learning. NeurIPS 2021
> > > [5] Zhou et al. OOD Link Prediction Generalization Capabilities of Message-Passing GNNs in Larger Test Graphs. arxiv:2205.15117

---

> > > > ### Comment · Reviewer_YzXG · 2022-08-08
> > > > **Better random guessing baselines**
> > > >
> > > > > Yes, the performance is non-trivial w.r.t random guessing - the probability of an answer to be in top-100 predictions over 2M nodes is 100 / 2*10^6  = 0.00005.
> > > >
> > > > I think the random baseline can be much stricter with answer type matching (entities that have an incoming edge of type *r* assuming the query has a type *r* edge leading into the answer node).
> > > >
> > > > Moreover, this can be made stronger with some kind of reordering by using the distance from anchor node. Entities of the expected answer type within a 3-hop neighborhood of the answer entity (I don't expect this will give much boost given your previous argument about graph diameter).
> > > >
> > > > The claim about non-trivial performance should be compared against such a stronger, non-trivial baseline.

---

> > > > > ### Author Response · Authors · 2022-08-09
> > > > > **WikiKG baseline**
> > > > >
> > > > > Thanks for the suggestion - the edge-type baseline should indeed be stronger.
> > > > > We finished implementing this baseline within those hours after your suggestion but did not manage to complete the experimental evaluation until the end of the discussion period. Nevertheless, we will add this baseline's results to both small and large datasets in the camera-ready version.

---

> > > ### Comment · Reviewer_YzXG · 2022-08-08
> > > **Clarification for initial increase in performance**
> > >
> > > > In simpler words, we hypothesize that the performance grows as long as the positive component outweighs the negative effect of OOD generalization
> > >
> > > Am I understanding correctly that the gain is due to the fact that the graph becomes sparser (average node degree decreases) as the graph grows? Can I look at a particular statistic in a table (from the paper/appendix) to see this in practice?

---

> > > > ### Author Response · Authors · 2022-08-08
> > > > **Response**
> > > >
> > > > Yes, this is correct. We provide dataset stats in Table 3 of Appendix B - average degree decreases from 27 to 5 in training graphs and from 40 to 20 in inference graphs.
> > > >
> > > > Please note that we included the whole Appendix into the revised manuscript here on OpenReview - once you open a PDF, it should include 10 new pages of the appendix after the main text and references. Appendix B is also available in the separate PDF in the zip archive with the supplementary materials.

---

> > ### Comment · Reviewer_YzXG · 2022-08-08
> > **Clarification on AUC**
> >
> > Thank you for thinking about this. I feel like this is an important aspect of the task that you are proposing. Do you plan to discuss this in the main paper?
> >
> > I do not see the updated appendix and cannot judge the section text. Can you post the text here?
> >
> > 1. I am missing some links here but I'm unsure how this is an "area under the curve". That is, I don't understand what curve leads to an area of 0.75 under it given your example.
> >
> > 2. Is this a sufficient metric for describing faithfulness? As you point out, this metric ignores the overall ranking quality. It does not account for predicting incorrect answers before the easy answers.
> >
> > 3. Suggestion: What if we consider the easy answers as the gold target and calculate hits@k? This would be for the evaluation queries as opposed to the train queries already reported. Do you see value/drawbacks in this metric?

---

> > > ### Author Response · Authors · 2022-08-08
> > > **Response**
> > >
> > > > Do you plan to discuss this in the main paper?
> > >
> > > Yes, camera-ready versions allow for one more page where we plan to include the ROC AUC discussion.
> > >
> > > > I do not see the updated appendix and cannot judge the section text. Can you post the text here?
> > >
> > > Sure, posted below. The text is also available in Appendix E - please note that we included the whole Appendix into the revised manuscript here on OpenReview - once you open a PDF, it should include 10 more pages of the appendix after the main text and references. All new materials are highlighted with the blue font.
> > >
> > >
> > > **Identifying Easy and Hard Answers**
> > >
> > > In addition to evaluating *faithfullness* that measures whether a model could recover easy answers, it is also interesting to measure whether all easy answers can be ranked higher than hard answers. That is, a reliable query answering model would first recover all possible easy answers and would enrich the answer set with highly-probable hard answers.
> > > To this end, we apply a ROC AUC metric over original *unfiltered* scores.
> > >
> > > The idea of computing ROC AUC is as follows:
> > > Suppose we have a list of unfiltered raw scores from which we extract scores of all easy and hard answers. Suppose we have a query with 4 easy and 1 hard answer: [5, 6, 7, 8, 32]  where 8 is a rank of a hard answer while 5, 6, 7, 32 are ranks of easy answers.
> > > We then create binary labels for the scores assigning 1 to the hard answers, e.g., [0, 0, 0, 1, 0].
> > >
> > > Given those two arrays, we then compute the ROC AUC score that would measure how many hard answers are ranked after easy answers, e.g., in our example ROC AUC is 0.75.
> > > Note that the score does not depend on actual values of ranks, that is, the metric will be high when easy answers are, e.g., ranked 1000-1004 as long as hard answers are ranked 1005 and lower.
> > > Therefore, ROC AUC still needs to be paired with MRR to see where easy and hard answers are ranked absolutely.
> > >
> > > We compute ROC AUC for each query and average them over each query type thus making it macro-averaged ROC AUC.
> > > Our experimental results on all query types using the models reported in Table 1 on the reference 175% dataset are compiled in Table 14.
> > >
> > > NBFNet-QE performs almost perfectly w.r.t. ROC AUC as it was trained on complex queries.
> > > NodePiece-QE models are acceptable for inference-only models that were only trained only on 1p simple link prediction and have never seen any complex query at training time.
> > >
> > > > That is, I don't understand what curve leads to an area of 0.75 under it given your example
> > >
> > > ROC AUC measures the probability of a random positive sample to have a higher rank than a random negative sample. In our example, we have 1 positive and 4 negative samples - choosing negatives randomly, we get 4 possible pairs (8 vs 5, 8 vs 6, 8 vs 7, 8 vs 32). In 3 out of 4 cases, the rank of positive sample 8 is higher, hence ROC AUC is also ¾ = 0.75.
> > >
> > > In simple words, ROC AUC measures how many data points with positive labels have higher scores (or higher ranks in our case) than data points with negative labels. In our case, positive labels are hard answers, and negative labels are easy answers. Having an array of binary labels and absolute scores (ranks), we can construct a ROC ([Receiver Operating Characteristic](https://en.wikipedia.org/wiki/Receiver_operating_characteristic)) curve (eg, using a [standard sklearn function](https://scikit-learn.org/stable/modules/generated/sklearn.metrics.roc_curve.html)) that depicts False Positive Rate on the X axis and True Positive Rate on the Y axis.
> > >
> > > > Is this a sufficient metric for describing faithfulness? What if we consider the easy answers as the gold target and calculate hits@k?
> > >
> > > ROC AUC does help to understand whether hard answers are ranked after easy ones (and it is a sufficient metric for that task) but we would argue that ROC AUC **alone** is not a sufficient metric to properly assess faithfulness - that is, we still need to combine it with filtered ranking metrics to assess absolute rank values.
> > >
> > > Generally, faithfulness as defined by Sun et al [1] is different from the ranking of easy answers before hard answers. Faithfulness is only about recovering easy answers in full graphs. Technically, we can rank all easy answers perfectly with a subgraph matching algorithm (which can’t solve any hard answer), it will be perfectly faithful but meaningless for real setting with graphs with missing links.
> > >
> > > We admit that a good query answering model generalizes to missing links and is faithful to existing links. However, it is not meaningful to look at the faithfulness alone, as a subgraph matching algorithm has perfect faithfulness but no generalization to missing links. We should also emphasize that faithfulness is much easier to achieve than generalization. For any neural model, we can ensemble it with a subgraph matching algorithm to achieve nearly perfect faithfulness.
> > >
> > > [1] Sun et al. Faithful Embedding for Knowledge Base Queries. NeurIPS 2020.

---

### Official Review · Reviewer_tzzT · 2022-07-13

**Rating:** 4
**Confidence:** 4
**Soundness:** 2 fair
**Presentation:** 3 good
**Contribution:** 2 fair

**Summary:**

The paper proposed a graph-based reasoning task in an inductive setting. Different from previously studied transductive reasoning task, a reasoning task contains unseen entities at inference time. Previous QE methods are unable to handle unseen entities.

**Questions:**

1. For both methods, do you make the assumption that "anchor" entities are seen at training time?

2. Could you please provide more information about the CQD-Beam method for decoding? How are the entity embeddings used in CQD-Beam? CQD-Beam is not a widely known methods so it may be confusing to readers.

3. In the NBFNet-QE method, the authors made an assumption that all edges (KB triples) that are required for prediction are present in the graph. How will your model be affected by an incomplete KB?


**Limitations:**

The authors claim the most important limitation of this work is the tradeoff between performance and computation complexity. While this is true, the authors should consider discussing more general limitations of this work, e.g. to what extent will this model work with unseen relations, sparse graphs with very few edges for nodes, types of supported reasoning operations, etc. More importantly, please compare your model with external baselines to justify your experiment results.

**Strengths And Weaknesses:**

The paper discussed two methods to handle unseen entities at inference time, NodePiece-QE and NBFNet-QE. In NodePiece-QE, the model computes embeddings for unseen entities from the relations of their neighbors. In NBFNet-QE, the model does not explicitly compute entity embeddings, but instead matching the graph structures locally. Both methods are plausible in solving the problem of novel entities. Experiment results show that the proposed methods achieve reasonable results.

However, I am a bit concern about the soundness of experiments. The authors experimented their models on a modified datasets to test models' ability in handle novel entities. They did not compare the performance of their model to any other external baselines, so it's not clear how well the proposed models perform. The reported metric, i.e. Hits@10, is highly dependent on the incoming / outgoing degree of the "anchor" nodes. Please consider carefully compare your models with other external baselines.

---

> ### Author Response · Authors · 2022-08-02
> **Response to Reviewer tzzT, Part 1**
>
> We thank the Reviewer for the valuable feedback and for acknowledging the plausibility of the proposed inductive models for complex query answering over unseen nodes.
>
> We would like to clarify the Reviewer's concerns about evaluation and comments on the limitations.
>
> >**W1. They did not compare the performance of their model to any other external baselines, so it's not clear how well the proposed models perform.**
>
> As you correctly mentioned in the summary, previous QE methods are unable to handle the inductive complex query answering task. To the best of our related work knowledge, there do not exist any inductive QE baselines with which we could directly compare NodePiece-QE and NBFNet-QE in the proposed task.
>
> That said, to further verify the effectiveness of inductive learning methods vs transductive ones, we trained BetaE [1], one of the most prominent state-of-the-art _transductive_ logical query answering models, with the convention that new, unseen entities in the inference graphs are initialized with random embeddings. It means that during training the model only updates the embeddings of training nodes (and some internal parameters), but at inference time it has to reason over both trained seen and randomly initialized unseen nodes.
> The BetaE configuration is configured similar to the best performing setup on transductive datasets including 400d embeddings and 200k training steps. The results on a reference 175% inductive split (reported in Table 1 & Section 5.2) are summarized below (and we also updated Table 1 in the manuscript and Appendix D with precise hyperparameters):
>
> |                      | avg\_p | avg\_n | 1p   | 2p   | 3p   | 2i   | 3i   | pi   | ip   | 2u   | up   | 2in  | 3in  | inp  | pin  | pni |
> | -------------------- | ------ | ------ | ---- | ---- | ---- | ---- | ---- | ---- | ---- | ---- | ---- | ---- | ---- | ---- | ---- | --- |
> | *BetaE (transductive)* | *1.8*    | *0.4*    | *2.8*  | *0.8*  | *0.4*  | *3.2*  | *5.1*  | *2.1*  | *1.1*  | *0.3*  | *0.3*  | *0.3*  | *0.7*  | *0.4*  | *0.3*  | *0.2* |
> | NodePiece-QE         | 11.3   | \-     | 24.1 | 10.5 | 10   | 11   | 12.8 | 9.1  | 8.6  | 7.3  | 8.3  | \-   | \-   | \-   | \-   | \-  |
> | NodePiece-QE w/ GNN  | 37.4   | \-     | 56.5 | 27.1 | 16   | 49.3 | 57.1 | 35.7 | 31.8 | 39.7 | 23.6 | \-   | \-   | \-   | \-   |     |
> | NBFNet-QE            | 51.1   | 31.4   | 66.1 | 40.9 | 31.2 | 73   | 83.3 | 58.3 | 41.3 | 37.8 | 27.8 | 31.1 | 44.3 | 28.4 | 25.2 | 28  |
>
> Clearly, a baseline transductive model does not demonstrate any reasonable performance on inductive tasks. The numbers, in turn, demonstrate that the proposed inductive baselines perform reasonably well and are non-trivial.
>
> As the crux of inductive reasoning in obtaining good representations of unseen nodes at inference time, our query answering framework might employ any existing QE approach known in transductive models as long as there is _some_ inductive representation mechanism. In this work, we paired NodePiece with CQD, an inference-only approach, to demonstrate the capabilities of inductive representation learning in the challenging inference-only setup, that is, when we do not train a model on complex queries at all. On the other hand, it is surely possible to pair NodePiece with BetaE (or anything else from the literature) where NodePiece will serve as an encoder learning inductive representations while BetaE (or anything else from the literature) will be performing query answering based on those representations.
>
> > **The reported metric, i.e. Hits@10, is highly dependent on the incoming / outgoing degree of the "anchor" nodes.**
>
> We would like to clarify that we compute Hits@k in the filtered setting - that is, if there are several true answers of a query, we mask out the scores of all the other true answers with (-inf) such that they do not interfere with the ranking of a target answer. Hence, the filtered Hits@k metric does not depend on incoming/outgoing degree. Besides, ranking metrics like MRR and Hits@k are rather standard in the query answering literature [1,2]. In the Appendix C, we also report a more challenging Hits@3 metric. Finally, following the question of Reviewer YzXG, we compute the ROC AUC score of easy vs hard answers of their original unfiltered scores showing that all proposed models are able to identify easy answers (via edge traversal) before hard answers (via link prediction).

---

> > ### Author Response · Authors · 2022-08-02
> > **Response to Reviewer tzzT, Part 2**
> >
> > > **Q1. For both methods, do you make the assumption that "anchor" entities are seen at training time?**
> >
> > No, “anchor” nodes might be totally new and unseen at inference time. In complex logical queries, anchor nodes are merely the starting entities from which a query is executed. At training time, the models only see the nodes from the training graph. At inference time (validation/test), when a graph is updated with many new nodes and edges, validation/test queries might start from those recently added nodes. In fact, in the first experiment reported in Section 5.2, we assume anchor entities of inference queries are new and unseen, and models have to predict missing links among unseen nodes; while in Section 5.3 we study the performance of original training queries (with anchors from the training graph) but executed on the larger inference graphs that add new answers to original training queries.
> >
> > > **Q2. Could you please provide more information about the CQD-Beam method for decoding? How are the entity embeddings used in CQD-Beam?**
> >
> > Apologies, the description didn’t make it to the main text due to space constraints - CQD [2] is a Continuous Query Decomposition method of answering complex queries with two variations - Continuous Optimization (CQD-CO) and Beam Search (CQD-Beam), one of the state-of-the-art query answering methods published at ICLR 2021. We employ CQD-Beam due to its inference-only non-parametric nature. CQD does not require training on actual complex queries and takes as input only entity and relation embedding matrices pre-trained on a simple link prediction task (1p in terms of complex queries) with a fixed scoring function. CQD is agnostic to the source of those pre-trained embeddings which allows us to replace transductively trained embeddings (as in the original paper) with inductive NodePiece representations (still pre-trained on 1p link prediction)
> >
> > Logical queries (with projection, intersection, and union) are decomposed into atomic *projection queries* (aka simple 1p link prediction that we solve by scoring each single triples with a scoring function) and non-parametric logical operations on top of them (t-norms and t-conorms that are essentially algebraic operations over two vectors). Projection queries are executed as a scoring function (ComplEx in our case) and induce a distribution of all entities in the current graph as possible answers. In queries that require multi-hop reasoning, we keep top-k scored entities as intermediate variables and execute further atomic queries using those top-k ranked nodes as starting nodes - this is essentially a Beam search well-known in natural language generation, hence the name CQD-Beam.
> >
> > We will extend the Section 4.1 with a more detailed description of CQD-Beam in the camera-ready version.
> >
> > > **Q3. In the NBFNet-QE method, the authors made an assumption that all edges (KB triples) that are required for prediction are present in the graph. How will your model be affected by an incomplete KB?**
> >
> > We might have chosen vague formulations - could you please point to the line in the manuscript where this assumption is made so we could clarify it? Generally, the task of complex query answering assumes the graph is incomplete by default and we have to predict missing links on the fly (at inference) to find *hard* answers. Both NodePiece-QE and NBFNet-QE are designed to work in incomplete graphs, so it’s totally fine.
> >
> > NBFNet-QE does indeed rely on the relational structure of a query (combinations of colored relations in Figure 2), but it does *not* require all edges to be physically present in the graph - in fact, the projection operator modeled with NBFNet [3] is able to predict those missing edges (or, to be more precise, predict a correct tail node of a query $(h, r, ?)$). This is achieved by the training procedure (similar to the original NBFNet link prediction objective) where, given an atomic projection query $(h, r, ?)$, we remove from the edge index all outgoing from $h$ edges of type $r$, and NBFNet has to learn to infer this edge of type $r$ outgoing from $h$ to possible intermediate or final answer nodes.

---

> > > ### Author Response · Authors · 2022-08-02
> > > **Response to Reviewer tzzT, Part 3**
> > >
> > > > **L1. .. consider discussing … to what extent will this model work with (i) unseen relations, (ii) sparse graphs with very few edges for nodes, (iii) types of supported reasoning operations**
> > >
> > > Thanks for bringing up those points!
> > >
> > > (i) Current inductive link prediction literature (e.g., GraIL [4], RED-GNN [5], NBFNet [3], and many more) assumes the set of relations is fixed and is not extended at inference time. We follow that formulation in Section 3 and assume that the set of relation types remains the same. The problem of inductive reasoning over *unseen relations* is highly complex and deserves a dedicated research paper - to the best of our knowledge, this setup has not been studied on theoretical or practical (like simple link prediction) levels in recent ML venues. This is definitely a solid avenue for a future work.
> > >
> > >
> > > (ii) GNNs, on which we base our models (NodePiece-QE w/ GNN employs CompGCN [6] and NBFNet-QE employs NBFNet, respectively) demonstrate a reasonably good performance on sparse graphs depending on chosen inductive biases. For example, NBFNet in the original paper shows high performance on inductive link prediction datasets where the average degree might be as low as 2. Our sampled datasets, in turn, vary in average degree from 5 to 37 (Appendix B).
> > >
> > >
> > > (iii) NodePiece-QE + CQD supports Existential Positive First Order (EPFO) queries - with projection, intersection, and union operators (lines 179-181 in the manuscript, Section 4.1). NBFNet-QE supports EPFO and queries with negation (lines 213-215, Section 4.2). Generally, inductive representation approaches (be it node or relational structure approaches) are quite orthogonal to the set of supported logical operators. That is, those approaches provide an *encoder* that yields representations, and it is a task of a *query decoder* to leverage those representations - whatever set of operators is supported by the query decoder, it would work in the inductive setup. That said, if CQD is extended to work with negation queries, NodePiece-QE + CQD will work with negation queries. It is also surely possible to swap CQD with another query answering decoder with its own set of logical operators.
> > >
> > > We will include that discussion in the camera-ready version as well.
> > >
> > > References:
> > >
> > > [1] Ren and Leskovec. Beta Embeddings for Multi-Hop Logical Reasoning in Knowledge Graphs. NeurIPS 2020.
> > > [2] Arakelyan et al. Complex Query Answering with Neural Link Predictors. ICLR 2021.
> > > [3] Zhu et al. Neural Bellman-Ford Networks: A General Graph Neural Network Framework for Link Prediction. NeurIPS 2021.
> > > [4] Teru et al. Inductive Relation Prediction by Subgraph Reasoning. ICML 2020
> > > [5] Zhang and Yao. Knowledge Graph Reasoning with Relational Digraph. WWW 2022
> > > [6] Vashishth et al. Composition-based Multi-Relational Graph Convolutional Networks. ICLR 2020.

---

> > ### Comment · Reviewer_YzXG · 2022-08-08
> > **Follow-up experiment to back the claim: Non-random initialization**
> >
> > > W1. They did not compare the performance of their model to any other external baselines, so it's not clear how well the proposed models perform.
> >
> > It's not surprising that random initialization of transductive baselines shows poor performance in inductive settings (thank you for demonstrating this). I feel to really drive the point home that simple modifications to transductive baselines will not work in this setting, authors can try to initialize embeddings for new entities smartly. e.g. embeddings for new entities can be initialized to minimize the 1p training objective while holding relation and old entity embeddings frozen.
> >
> > This would further strengthen the original claim that the proposed approaches have strong performance without training on growing graphs.
> >
> > This may require fine-tuning for BetaE but might be easier to do for transductive link prediction models that are based on geometry like RotatE (new entity embeddings are the average of rotated embeddings of neighbors).

---

> > > ### Author Response · Authors · 2022-08-08
> > > **Response**
> > >
> > > As we are at the end of the discussion period, we won’t be able to add new experiments to the current revision. Still, we would argue that any fine-tuning of embeddings of new nodes at inference time essentially leads to:
> > >
> > > (1) either creating a new inductive model - like averaging of neighboring training entities - this actually has been proposed in [1] and is likely to work only in relatively dense graph scenarios when the number of new nodes is much less than the number of known trained nodes. That is, for example, in our scenarios of up to 550% of new nodes, most new nodes will be aggregating randomly initialized vectors and the signal of few trained embeddings is likely to disappear in the noise. In fact, any node neighborhood aggregation is akin to message passing and we have shown in the original experiments with NodePiece-QE and NBFNet-QE that message passing GNNs do help to enrich node representations but still deteriorate with the growth of the inference graph.
> > >
> > > Or (2) re-training node embeddings over the inference graph - that is the transductive scenario again with all nodes known (like the proposed task of additional link prediction training after adding new nodes). Particularly in scenarios with large inference graphs, like that of 217% and higher, where the inference graph is 2.5-25x times larger in the number of edges, any additional fine-tuning / optimization would take much more time than original training. Practically, this step does not conform with the goal of learning inductive representations and fast inference which we can achieve in one model forward pass - order of (milli)seconds - whereas re-training every time the graph changes costs hours.
> > >
> > > Both options mentioned by the Reviewer differ from the standard inductive setup (eg, defined in the GraIL paper [2] where new nodes in the inference graph arrive only with the connectivity information without any input node features) where we are not allowed such fine-tuning and we would like to obtain inductive node / relational structure representations in the true inductive zero-shot way.
> > >
> > > [1] Albooyeh et al. Out-of-Sample Representation Learning for Knowledge Graphs. EMNLP Findings 2020
> > > [2] Teru et al. Inductive Relation Prediction by Subgraph Reasoning. ICML 2020.

---

### Author Response · Authors · 2022-08-02
**General Response**

We thank the reviewers for the valuable feedback that helped to improve the manuscript. In addition to particular responses, in this general response, we would like to draw your attention to major manuscript updates (marked in the blue color in the new version) and 4 new experimental results.

1. Addressing the comments of Reviewer **YzXG** and **YfZT**, we updated all results of NodePiece-QE and NodePiece w/ GNN thanks to the improved pre-training strategy. In particular, for the GNN version, it leads to almost 2X better performance in both test queries (Table 1, Figure 3 in the main text, and Appendix C) and 1.5X on training queries (Figure 4 in the main text, and Appendix C) as a direct outcome of better fitting the training data. With that, the gap between the inference-only NodePiece-QE w/ GNN and trainable NBFNet-QE is significantly reduced, and in some queries and dataset ratios, the inference-only model even outperforms the trainable model, e.g., in *ip* and *2u* queries (Figure 5 in Appendix C). We believe this is a strong signal that better fitting of a simple link prediction model might result in significant performance gains.

2. Addressing the comments of Reviewer **tzzT**, we experimented with transductive BetaE where embeddings of new nodes at inference time are initialized randomly. BetaE results on the reference 175% dataset are attached below and added to Table 1 with hyperparameter description in Appendix D. The results suggest that state-of-the-art transductive complex query answering approaches do not work in the inductive setup.

|                      | avg\_p | avg\_n | 1p   | 2p   | 3p   | 2i   | 3i   | pi   | ip   | 2u   | up   | 2in  | 3in  | inp  | pin  | pni |
| -------------------- | ------ | ------ | ---- | ---- | ---- | ---- | ---- | ---- | ---- | ---- | ---- | ---- | ---- | ---- | ---- | --- |
| *BetaE (transductive)* | *1.8*    | *0.4*    | *2.8*  | *0.8*  | *0.4*  | *3.2*  | *5.1*  | *2.1*  | *1.1*  | *0.3*  | *0.3*  | *0.3*  | *0.7*  | *0.4*  | *0.3*  | *0.2* |
| NodePiece-QE         | 11.3   | \-     | 24.1 | 10.5 | 10   | 11   | 12.8 | 9.1  | 8.6  | 7.3  | 8.3  | \-   | \-   | \-   | \-   | \-  |
| NodePiece-QE w/ GNN  | 37.4   | \-     | 56.5 | 27.1 | 16   | 49.3 | 57.1 | 35.7 | 31.8 | 39.7 | 23.6 | \-   | \-   | \-   | \-   |     |
| NBFNet-QE            | 51.1   | 31.4   | 66.1 | 40.9 | 31.2 | 73   | 83.3 | 58.3 | 41.3 | 37.8 | 27.8 | 31.1 | 44.3 | 28.4 | 25.2 | 28  |

3. Addressing the comments of Reviewer **YzXG**, we ran an experiment to identify whether all “easy” answers (not requiring link prediction) are ranked higher than “hard” answers (that do require inferring missing edges). For that, we compute **macro-averaged ROC AUC** score from **unfiltered** predictions. Results on the reference 175% dataset are presented below and added to Appendix E with further details on the methodology and setup. We will add ROC AUC results for other splits in the final version.

|                     | avg_p  | avg_n   | 1p    | 2p    | 3p    | 2i    | 3i    | pi    | ip    | 2u    | up    | 2in   | 3in   | inp   | pin   | pni   |
| ------------------- | ----- | ----- | ----- | ----- | ----- | ----- | ----- | ----- | ----- | ----- | ----- | ----- | ----- | ----- | ----- | ----- |
| NodePiece-QE        | 0.653 |       | 0.609 | 0.656 | 0.666 | 0.628 | 0.627 | 0.645 | 0.681 | 0.677 | 0.686 |       |       |       |       |       |
| NodePiece-QE w/ GNN | 0.675 |       | 0.688 | 0.689 | 0.667 | 0.655 | 0.624 | 0.646 | 0.69  | 0.713 | 0.701 |       |       |       |       |       |
| NBFNet-QE           | 0.978 | 0.901 | 0.997 | 0.981 | 0.962 | 0.987 | 0.978 | 0.975 | 0.975 | 0.982 | 0.965 | 0.919 | 0.884 | 0.888 | 0.913 | 0.904 |

The results suggest that NBFNet-QE performs almost perfectly w.r.t. ROC AUC as it was trained on complex queries. NodePiece-QE models are pretty good for inference-only models that were only trained only on 1p simple link prediction and have never seen any complex query at training time.

4. Addressing the comments of Reviewers **YzXG** and **YfZT**, we updated Table 2 with the performance of NodePiece-QE on the large WikiKG dataset increasing the average EPFO Hits@100 performance by relative 40% (from 6.6 to 9.2) and also made an attempt to train NodePiece-QE w/ GNN in the full-batch mode for 1 epoch. GNN results are slightly higher on average but that is achieved mainly by 1p accuracy. We will run a GNN pre-training for a longer time and update the results in the final version.

|                     | EPFO | 1p   | 2p  | 3p  | 2i   | 3i   | pi  | ip  | 2u  | up  |
| ------------------- | ---- | ---- | --- | --- | ---- | ---- | --- | --- | --- | --- |
| NodePiece-QE        | 9.2  | 22.6 | 5.2 | 3.9 | 11.6 | 17.4 | 7   | 4.5 | 7.4 | 3.2 |
| NodePiece-QE w/ GNN | 10.1 | 66.6 | 0.9 | 0.6 | 5.4  | 8.2  | 2.3 | 0.8 | 5.2 | 0.5 |


Thank you and please feel free to ask any further question.

---

### Meta-Review · Area_Chair_vkQZ · 2022-08-26

**Recommendation:** Accept
**Confidence:** Less certain

**Metareview:**

It is very hard to make a final decision on this paper; the scores are: 4,6,8, and 5. The research problem raised in this paper is interesting and worth further study.
However, reviewers have raised some concerns about the experimental resuls. In the original paper, they did not compare with any external baselines. In the discussion period, the authors presented additional results with BetaE, which were originally designed for transductive reasoning. Not surprisingly, BetaE delivers poor results. How strong is this comparison? One would expect some experiments with better initialization for unseen entities, e.g. with KB embeddings (e.g. TransE and ConvE used in KBC tasks).

**Award:**

No

---

### Decision · Program_Chairs · 2022-09-14

Accept